# Non-asymptotic Analysis of Stochastic Methods for Non-Smooth Non-Convex Regularized Problems

**Yi Xu**[1]**, Rong Jin**[2]**, Tianbao Yang**[1]

1. Department of Computer Science, The University of Iowa, Iowa City, IA 52246, USA
2. Machine Intelligence Technology, Alibaba Group, Bellevue, WA 98004, USA
`{yi-xu, tianbao-yang}@uiowa.edu, jinrong.jr@alibaba-inc.com`

## Abstract

**S**tochastic **P**roximal **G**radient (**SPG**) methods have been widely used for solving optimization problems with a simple (possibly non-smooth) regularizer in machine learning and statistics. However, to the best of our knowledge no non-asymptotic convergence analysis of SPG exists for non-convex optimization with a **non-smooth and non-convex regularizer**. All existing non-asymptotic analysis of SPG for solving non-smooth non-convex problems require the non-smooth regularizer to be a convex function, and hence are not applicable to a non-smooth non-convex regularized problem. This work initiates the analysis to bridge this gap and opens the door to non-asymptotic convergence analysis of non-smooth non-convex regularized problems. We analyze several variants of **mini-batch** SPG methods for minimizing a non-convex objective that consists of a smooth non-convex loss and a non-smooth non-convex regularizer. Our contributions are two-fold: (i) we show that they enjoy the same complexities as their counterparts for solving convex regularized non-convex problems in terms of finding an approximate stationary point; (ii) we develop more practical variants using dynamic mini-batch size instead of a fixed mini-batch size without requiring the target accuracy level of solution. The significance of our results is that they improve upon the-state-of-art results for solving non-smooth non-convex regularized problems. We also empirically demonstrate the effectiveness of the considered SPG methods in comparison with other peer stochastic methods.

## 1 Introduction

In this work, we consider the following stochastic non-smooth non-convex optimization problem:

$$\min_{\mathbf{x} \in \mathbb{R}^d} F(\mathbf{x}) := \underbrace{\mathrm{E}_{\xi}[f(\mathbf{x}; \xi)]}_{f(\mathbf{x})} + r(\mathbf{x}), \tag{1}$$

where $\xi$ is a random variable, $f(\mathbf{x})$ is a smooth non-convex function, and $r(\mathbf{x}) : \mathbb{R}^d \to \mathbb{R}$ is a proper non-smooth non-convex lower-semicontinuous function. A special case of problem (1) in machine learning is of the following finite-sum form:

$$\min_{\mathbf{x} \in \mathbb{R}^d} F(\mathbf{x}) := \frac{1}{n} \sum_{i=1}^{n} f_i(\mathbf{x}) + r(\mathbf{x}), \tag{2}$$

where $n$ is the number of data samples. In the sequel, we refer to the problem (1) with a finite-sum structure as in the finite-sum setting and otherwise as in the online setting [29, 43]. The family of optimization problems with a non-convex smooth loss and a non-convex non-smooth regularizer is important and broad in machine learning and statistics. Examples of smooth non-convex losses include non-linear square loss for classification [20], truncated square loss for regression [44], and cross-entropy loss for learning a neural network with a smooth activation function. Examples of

Table 1: Summary of Complexities for finding an $\epsilon$-stationary point of (1). LC denotes Lipchitz continuous function; FV means finite-valued over $\mathbb{R}^d$; PM denotes the proximal mapping exists and can be obtained efficiently. $\widetilde{O}(\cdot)$ suppresses a logarithmic factor in terms of $\epsilon^{-1}$.

| Problem | Algorithm | complexity | $r(\mathbf{x})$ |
|---|---|---|---|
| Online | MBSGA [29], SSDC-SPG [43] | $O(\epsilon^{-5})$ | PM, LC |
| Online | SSDC-SPG [43] | $O(\epsilon^{-6})$ | PM, FV |
| Online | **MB-SPG (this work)** | $O(\epsilon^{-4})$ | PM |
| Online | **SPGR (this work)** | $O(\epsilon^{-3})$ | PM |
| Finite-sum | VRSGA [29] | $O(n^{2/3}\epsilon^{-3})$ | PM, LC |
| Finite-sum | SSDC-SVRG [43] | $\widetilde{O}(n\epsilon^{-3})$ | PM, LC |
| Finite-sum | SSDC-SVRG [43] | $\widetilde{O}(n\epsilon^{-4})$ | PM, FV |
| Finite-sum | **SPGR (this work)** | $O(n^{1/2}\epsilon^{-2} + n)$ | PM |

non-smooth non-convex regualerizers include $\ell_p$ ($0 \le p < 1$) norm, smoothly clipped absolute deviation (SCAD) [17], log-sum penalty (LSP) [9], minimax concave penalty (MCP) [48], and an indicator function of a non-convex constraint as well (e.g., $\|\mathbf{x}\|_0 \le k$).

Although non-convex minimization with a non-smooth convex regularizer has been extensively studied in both online setting [19, 14, 41, 35] and finite-sum setting [16, 38, 1, 34, 26, 12, 41, 35], stochastic optimization for the considered problem with a non-smooth non-convex regularizer is still under-explored. The presence of non-smooth non-convex functions $r$ makes the analysis more challenging, which renders previous analysis that hinges on the convexity of $r$ not applicable. A special case of non-convex $r$ that can be written as a DC (Difference of Convex) function, i.e., $r(\mathbf{x}) = r_1(\mathbf{x}) - r_2(\mathbf{x})$ with $r_1$ and $r_2$ being convex, has been recently tackled by several studies with stochastic algorithms [43, 33, 40]. In this paper, we focus on first-order stochastic algorithms for solving the problem (1) with a general non-smooth non-convex regularizer and study their non-asymptotic convergence rates.

Although there are plenty of studies devoted to non-smooth non-convex regularized problems [3, 5, 49, 23, 25, 47, 6, 2, 46, 27], they are restricted to deterministic algorithms and asymptotic or local convergence analysis. There are *few* studies concerned with the non-asymptotic convergence analysis of stochastic algorithms for the problem (1). To the best of our knowledge, [43] is the first work that presents stochastic algorithms with non-asymptotic convergence results for finding an approximate critical point of a non-convex problem with a non-convex non-smooth regularizer. Indeed, they considered a more general problem in which $f$ is a DC function and assumed that the second component of the DC decomposition of $f$ has a Hölder-continuous gradient. Their convergence results are the state-of-the-art for stochastic optimization of the problem (1) in the online setting. Later, [29] presented two algorithms, namely mini-batch stochastic gradient algorithm (MBSGA) and variance reduced stochastic gradient algorithm (VRSGA), for solving (1) and (2) with an improved complexity for the finite-sum setting. To tackle the non-smooth non-convex regularizer, both of these works use a Moreau envelope of $r$ to approximate $r$, which inevitably introduces approximation error and hence worsen the convergence rates.

A simple idea for tackling a non-smooth regularizer is to use proximal gradient methods, which has been studied extensively in the literature for a convex regularizer [19, 14, 16, 38, 1, 34, 26, 12, 41, 35]. A natural question is whether stochastic proximal gradient (SPG) methods still enjoy similar convergence guarantee for solving a non-smooth non-convex regularized problem as their counterparts for convex regularized non-convex minimization problems. In this paper, we provide an affirmative answer to this question. Our contributions are summarized below:

- We establish **the first convergence rate** of standard mini-batch SPG (MB-SPG) for solving (1) in terms of finding an approximate stationary point, which is the same as its counterpart for solving a non-convex minimization problem with a convex regularizer [19].

- Furthermore, we analyze improved variants of mini-batch SPG that use a recursive stochastic gradient estimator (SARAH [32, 31] or SPIDER [18, 41]) referred to as SPGR, and achieve **the new state of the art** convergence results for both online setting and the finite-sum setting.

- Moreover, we propose **more practical** variants of MB-SPG and SPGR by using dynamic mini-batch size instead of a fixed mini-batch size to remove the requirement on the target accuracy level of solution for running the algorithms.

The complexity results of our algorithms and other works for finding an $\epsilon$-stationary solution of the considered problem are summarized in Table 1. It is notable that the complexity result of SPGR for the finite-sum setting is optimal matching an existing lower bound [18]. Before ending this section, it is worth mentioning that the differences between this work and [15] that provides the first convergence analysis of SPG to critical points of a non-smooth non-convex minimization problem: (i) their convergence analysis is asymptotic and hence provides no convergence rate; (ii) their analysis applies to non-smooth $f$ but requires stronger assumptions on $r$ (e.g., local Lipchitz continuity) that precludes $\ell_0$ norm regularizer or an indicator function of a non-convex constraint; (ii) their analyzed SPG imposes no requirement on the mini-batch size.

## 2 Preliminaries

In this section, we present some preliminaries and notations. Let $\|\mathbf{x}\|$ denote the Euclidean norm of a vector $\mathbf{x} \in \mathbb{R}^d$. Denote by $\mathcal{S} = \{\xi_1, \ldots, \xi_m\}$ a set of random variables, let $|\mathcal{S}|$ be the number of elements in set $\mathcal{S}$ and $f_{\mathcal{S}}(\mathbf{x}) = \frac{1}{|\mathcal{S}|} \sum_{\xi_i \in \mathcal{S}} f(\mathbf{x}; \xi_i)$. We denote by $\text{dist}(\mathbf{x}, \mathcal{S})$ the distance between the vector $\mathbf{x}$ and a set $\mathcal{S}$. Denote by $\hat{\partial}h(\mathbf{x})$ the Fréchet sub-gradient and $\partial h(\mathbf{x})$ the limiting subgradient of a non-convex function $h(\mathbf{x}) : \mathbb{R}^d \to \mathbb{R}$, i.e., $\hat{\partial}h(\bar{\mathbf{x}}) = \left\{ \mathbf{v} \in \mathbb{R}^d : \lim_{\mathbf{x} \to \bar{\mathbf{x}}} \inf \frac{h(\mathbf{x}) - h(\bar{\mathbf{x}}) - \mathbf{v}^\top (\mathbf{x} - \bar{\mathbf{x}})}{\|\mathbf{x} - \bar{\mathbf{x}}\|} \geq 0 \right\}, \partial h(\bar{\mathbf{x}}) = \{ \mathbf{v} \in \mathbb{R}^d : \exists \mathbf{x}_k \xrightarrow{h} \bar{\mathbf{x}}, v_k \in \hat{\partial}h(\mathbf{x}_k), \mathbf{v}_k \to \mathbf{v} \}$, where $\mathbf{x} \xrightarrow{h} \bar{\mathbf{x}}$ means $\mathbf{x} \to \bar{\mathbf{x}}$ and $h(\mathbf{x}) \to h(\bar{\mathbf{x}})$.

We aim to find an $\epsilon$-stationary point of problem (1), i.e., to find a solution $\mathbf{x}$ such that $\text{dist}(0, \hat{\partial}F(\mathbf{x})) \leq \epsilon$. Since $f$ is differentiable, then we have $\hat{\partial}F(\mathbf{x}) = \hat{\partial}(f + r)(\mathbf{x}) = \nabla f(\mathbf{x}) + \hat{\partial}r(\mathbf{x})$ (see Exercise 8.8, [39]). Thus, it is equivalent to find a solution $\mathbf{x}$ satisfying

$$\text{dist}(0, \nabla f(\mathbf{x}) + \hat{\partial}r(\mathbf{x})) \leq \epsilon. \tag{3}$$

For problem (1), we make the following basic assumptions, which are standard in the literature on stochastic gradient methods for non-convex optimization [19, 29].

**Assumption 1.** *Assume the following conditions hold:*

*(i)* $\mathrm{E}_\xi[\nabla f(\mathbf{x}; \xi)] = \nabla f(\mathbf{x})$, *and there exists a constant* $\sigma > 0$, *such that* $\mathrm{E}_\xi[\|\nabla f(\mathbf{x}; \xi) - \nabla f(\mathbf{x})\|^2] \leq \sigma^2$.

*(ii)* *Given an initial point* $\mathbf{x}_0$, *there exists* $\Delta < \infty$ *such that* $F(\mathbf{x}_0) - F(\mathbf{x}_*) \leq \Delta$, *where* $\mathbf{x}_*$ *denotes the global minimum of (1).*

*(iii)* $f(\mathbf{x})$ *is smooth with a L-Lipchitz continuous gradient, i.e., it is differentiable and there exists a constant* $L > 0$ *such that* $\|\nabla f(\mathbf{x}) - \nabla f(\mathbf{y})\| \leq L\|\mathbf{x} - \mathbf{y}\|, \forall \mathbf{x}, \mathbf{y}$.

In addition, we assume $r(\mathbf{x})$ is simple enough such that its proximal mapping exists and can be obtained efficiently:

$$\text{prox}_{\eta r}[\mathbf{x}] = \arg\min_{\mathbf{y} \in \mathbb{R}^d} \left\{ \frac{1}{2\eta}\|\mathbf{y} - \mathbf{x}\|^2 + r(\mathbf{y}) \right\}.$$

This assumption is standard to proximal algorithms for non-convex functions [3, 8, 24]. The notation $\arg\min$ denotes the set of minimizers. The closed form of proximal mapping for non-convex regularizers include hard thresholding for $\ell_0$ regularizer [3], and $\ell_p$ thresholding for $\ell_{1/2}$ regularizer [45] and $\ell_{2/3}$ regularizer [10].

An immediate difficulty in solving problem (1) is the presence of non-smoothness non-convexity in the regularizer $r(\mathbf{x})$. To deal with this issue, [43, 29] use the the Moreau envelope of $r$ to approximate $r$, which is defined as $r_\mu(\mathbf{x}) = \min_{\mathbf{y} \in \mathbb{R}^d} \left\{ \frac{1}{2\mu}\|\mathbf{y} - \mathbf{x}\|^2 + r(\mathbf{y}) \right\}$, where $\mu > 0$ is an approximation parameter. It is easy to see that the Moreau envelope of $r(\mathbf{x})$ is a DC function:

$$r_\mu(\mathbf{x}) = \frac{1}{2\mu}\|\mathbf{x}\|^2 - \underbrace{\max_{\mathbf{y} \in \mathbb{R}^d} \frac{1}{\mu}\mathbf{y}^\top \mathbf{x} - \frac{1}{2\mu}\|\mathbf{y}\|^2 - r(\mathbf{y})}_{R_\mu(\mathbf{x})},$$

where $R_\mu(\mathbf{x})$ is convex since it is the max of convex functions in terms of $\mathbf{x}$ [7]. Instead of solving the problem (1) directly, their idea is to solve the following approximated problem:

$$\min_{\mathbf{x}\in\mathbb{R}^d} F_\mu(\mathbf{x}) := f(\mathbf{x}) + \frac{1}{2\mu}\|\mathbf{x}\|^2 - R_\mu(\mathbf{x}). \qquad (4)$$

However, this is a bad idea because it introduces the approximation error on one hand and slows down the convergence on the other hand. For example, [29] considers algorithms that update the solution based on a smooth function that is constructed by linearizing the term $R_\mu(\mathbf{x})$. As a result, the smoothness constant of the resulting function is proportional to $1/\mu$. In order to maintain a small approximation error, $\mu$ has to be a small value which amplifies the smoothness constant dramatically.

In this paper, we consider a direct approach that updates the solution simply by a stochastic proximal gradient update, i.e., $\mathbf{x}_{t+1} \in \text{prox}_{\eta r}[\mathbf{x}_t - \eta \mathbf{g}_t]$, where $\mathbf{g}_t$ is a stochastic gradient of $\nabla f(\mathbf{x}_t)$ with well-controlled variance, and $\eta$ is a step size.

## 2.1 Warm-up: Proximal Gradient Descent Method

As a warm-up, we first present the analysis of the deterministic proximal gradient descent (PGD) method (also known as forward-backward splitting, FBS), which updates the solutions for $t = 0, \ldots, T-1$ iteratively given an initial solution $\mathbf{x}_0$:

$$\mathbf{x}_{t+1} \in \text{prox}_{\eta r}[\mathbf{x}_t - \eta\nabla f(\mathbf{x}_t)] = \arg\min_{\mathbf{x}\in\mathbb{R}^d}\left\{ r(\mathbf{x}) + \langle\nabla f(\mathbf{x}_t), \mathbf{x} - \mathbf{x}_t\rangle + \frac{1}{2\eta}\|\mathbf{x} - \mathbf{x}_t\|^2 \right\}, \quad (5)$$

where $\eta$ is a step size. To our knowledge, non-asymptotic analysis of PGD for non-convex $r(\mathbf{x})$ is not available, though asymptotic analysis of PGD was provided in [3]. We summarize the non-asymptotic convergence result of PGD in the following theorem, and provide a proof sketch to highlight the key steps. The detailed proofs are provided in the supplement.

**Theorem 1.** *Suppose Assumption 1 (ii) and (iii) hold, run (5) with $\eta = \frac{c}{L}$ ($0 < c < 1$) and $T = \frac{4(\eta^2 L^2 + 1)}{\eta(1-\eta L)\epsilon^2}\Delta = O(1/\epsilon^2)$ iterations, with $R$ being uniformly sampled from $\{1, \ldots, T\}$ we have $\mathrm{E}[dist(0, \hat\partial F(\mathbf{x}_R))] \leq \epsilon$.*

**Remark:** It is notable that this complexity result is optimal according to [11] for smooth non-convex optimization, which is the same as that for solving problem (1) when $r(\mathbf{x})$ is convex [30].

*Proof Sketch.* For the update (5), we can only leverage its optimality condition (e.g., by Exercise 8.8 and Theorem 10.1 of [39]):

$$-\nabla f(\mathbf{x}_t) - \frac{1}{\eta}(\mathbf{x}_{t+1} - \mathbf{x}_t) \in \hat\partial r(\mathbf{x}_{t+1}),$$

$$r(\mathbf{x}_{t+1}) + \langle\nabla f(\mathbf{x}_t), \mathbf{x}_{t+1} - \mathbf{x}_t\rangle + \frac{1}{2\eta}\|\mathbf{x}_{t+1} - \mathbf{x}_t\|^2 \leq r(\mathbf{x}_t),$$

where the first implies that $\nabla f(\mathbf{x}_{t+1}) - \nabla f(\mathbf{x}_t) - \frac{1}{\eta}(\mathbf{x}_{t+1} - \mathbf{x}_t) \in \hat\partial F(\mathbf{x}_{t+1})$. Combining the second inequality with the smoothness of $f(\mathbf{x})$, i.e., $f(\mathbf{x}_{t+1}) \leq f(\mathbf{x}_t) + \langle\nabla f(\mathbf{x}_t), \mathbf{x}_{t+1} - \mathbf{x}_t\rangle + \frac{L}{2}\|\mathbf{x}_{t+1} - \mathbf{x}_t\|^2$, we get $\frac{1}{2}(1/\eta - L)\|\mathbf{x}_{t+1} - \mathbf{x}_t\|^2 \leq F(\mathbf{x}_t) - F(\mathbf{x}_{t+1})$. By telescoping the above inequality and connecting $\hat\partial F(\mathbf{x}_{t+1})$ with $\|\mathbf{x}_{t+1} - \mathbf{x}_t\|$ we can finish the proof. $\qquad\square$

## 3 Mini-Batch Stochastic Proximal Gradient Methods

In this and next section, we analyze mini-batch stochastic proximal gradient methods that use a stochastic gradient $\mathbf{g}_t$ for updating the solution. The key idea of the two methods is to control the variance of the stochastic gradient properly.

We present the detailed updates of the first algorithm (named MB-SPG) in Algorithm 1, which is to update the solution based on a mini-batched stochastic gradient of $f(\mathbf{x})$ at the $t$-th iteration and the proximal mapping of $r(\mathbf{x})$. We first present a general convergence result of Algorithm 1.

**Theorem 2.** *Suppose Assumption 1 holds, run Algorithm 1 with $\eta = \frac{c}{L}$ ($0 < c < \frac{1}{2}$), then the output $\mathbf{x}_R$ of Algorithm 1 satisfies $\mathrm{E}[dist(0, \hat\partial F(\mathbf{x}_R))^2] \leq \frac{c_1}{T}\sum_{t=0}^{T-1} \mathrm{E}[\|\mathbf{g}_t - \nabla f(\mathbf{x}_t)\|^2] + \frac{c_2\Delta}{\eta T}$, where $c_1 = \frac{2c(1-2c)+2}{c(1-2c)}$ and $c_2 = \frac{6-4c}{1-2c}$ are two positive constants.*

---
**Algorithm 1** Mini-Batch Stochastic Proximal Gradient: MB-SPG
---
1: **Initialize**: $\mathbf{x}_0 \in \mathbb{R}^d$, $\eta = \frac{c}{L}$ with $0 < c < \frac{1}{2}$.
2: **for** $t = 0, 1, \ldots, T - 1$ **do**
3:     Draw samples $\mathcal{S}_t = \{\xi_i, \ldots, \xi_{m_t}\}$, let $\mathbf{g}_t = \frac{1}{m_t} \sum_{i_t=1}^{m_t} \nabla f(\mathbf{x}_t; \xi_{i_t})$
4:     $\mathbf{x}_{t+1} \in \text{prox}_{\eta r}[\mathbf{x}_t - \eta \mathbf{g}_t]$
5: **end for**
6: **Output:** $\mathbf{x}_R$, where $R$ is uniformly sampled from $\{1, \ldots, T\}$.
---

---
**Algorithm 2** Stochastic Proximal Gradient using SPIDER/SARAH: SPGR
---
1: **Initialize**: $\mathbf{x}_0 \in \mathbb{R}^d$, $\eta = \frac{c}{L}$ with $0 < c < \frac{1}{3}$.
2: **for** $t = 0, 1, \ldots, T - 1$ **do**
3:     **if** $\text{mod}(t, q) == 0$ **then**
4:        Draw samples $\mathcal{S}_1$, let $\mathbf{g}_t = \nabla f_{\mathcal{S}_1}(\mathbf{x}_t)$ // For finite-sum setting, $|\mathcal{S}_1| = n$
5:     **else**
6:        Draw samples $\mathcal{S}_2$, let $\mathbf{g}_t = \nabla f_{\mathcal{S}_2}(\mathbf{x}_t) - \nabla f_{\mathcal{S}_2}(\mathbf{x}_{t-1}) + \mathbf{g}_{t-1}$
7:     **end if**
8:     $\mathbf{x}_{t+1} \in \text{prox}_{\eta r}[\mathbf{x}_t - \eta \mathbf{g}_t]$
9: **end for**
10: **Output:** $\mathbf{x}_R$, where $R$ is uniformly sampled from $\{1, \ldots, T\}$.
---

Next, we present two corollaries by using a fixed mini-batch size and increasing mini-batch sizes.

**Corollary 3** (Fixed mini-batch size)**.** *Suppose Assumption 1 holds, run MB-SPG (Algorithm 1) with $\eta = \frac{c}{L}$ ($0 < c < \frac{1}{2}$), $T = 2c_2 \Delta / (\eta \epsilon^2)$ and a fixed mini-batch size $m_t = 2c_1 \sigma^2 / \epsilon^2$ for $t = 0, \ldots, T - 1$, then the output $\mathbf{x}_R$ of Algorithm 1 satisfies $\mathrm{E}[dist(0, \hat{\partial} F(\mathbf{x}_R))^2] \leq \epsilon^2$, where $c_1, c_2$ are two positive constants as in Theorem 2.*

**Corollary 4** (Increasing mini-batch sizes)**.** *Suppose Assumption 1 holds, run MB-SPG (Algorithm 1) with $\eta = \frac{c}{L}$ ($0 < c < \frac{1}{2}$) and a sequence of mini-batch sizes $m_t = b(t + 1)$ for $t = 0, \ldots, T - 1$, where $b > 0$ is a constant, then the output $\mathbf{x}_R$ of Algorithm 1 satisfies $\mathrm{E}[dist(0, \hat{\partial} F(\mathbf{x}_R))^2] \leq \frac{c_1 \sigma^2 (\log(T) + 1)}{bT} + \frac{c_2 \Delta}{\eta T}$, where $c_1, c_2$ are constants as in Theorem 2. In particular in order to have $\mathrm{E}[dist(0, \hat{\partial} F(\mathbf{x}_R))] \leq \epsilon$, it suffices to set $T = \widetilde{O}(1/\epsilon^2)$. The total complexity is $\widetilde{O}(1/\epsilon^4)$.*

**Remark:** Although using increasing mini-batch sizes has an additional logarithmic factor in the complexity than that using a fixed mini-batch size, it would be more practical and user-friendly because it does not require knowing the target accuracy $\epsilon$ to run the algorithm .

## 4 Stochastic Proximal Gradient Methods with Recursive Stochastic Gradient Estimator

In this section, we leverearge the novel recursive stochastic gradient estimator (SARAH/SPIDER) for achieving a better complexity. We present the detailed updates of the proposed algorithm referred to as SPGR in Algorithm 2, where the stochastic gradient estimate $\mathbf{g}_t$ is periodically updated by adding current stochastic gradient $\nabla f_{\mathcal{S}_2}(\mathbf{x}_t)$ and subtracting the past stochastic gradient $\nabla f_{\mathcal{S}_2}(\mathbf{x}_{t-1})$ from $\mathbf{g}_{t-1}$. To our knowledge, this framework was firstly introduced in SARAH [32, 31] for solving convex/nonconvex smooth finite-sum problems with $r(\mathbf{x}) = 0$. Another algorithm so-called SPIDER with same recursive framework was proposed in [18] for solving non-convex smooth problems with $r(\mathbf{x}) = 0$ both in finite-sum and online settings. One difference is that SPIDER uses normalized gradient update with step size $\eta = O(\epsilon/L)$. Recently, [41] and [35] respectively extended SPIDER and SARAH to their proximal versions for solving non-convex smooth problems with convex non-smooth regularizer $r(\mathbf{x})$. By contrast, we consider more challenging problems in this paper, i.e., non-convex non-smooth regularized non-convex minimization problems. In order to use the SARAH/SPIDER technique to construct a variance-reduced stochastic gradient of $f$, we need additional assumption, which is also used in previous studies [31, 18, 41, 35].

**Assumption 2.** *Assume that every random function $f(\mathbf{x}; \xi)$ is smooth with a L-Lipchitz continuous gradient, i.e., it is differentiable and there exists a constant $L > 0$ such that $\|\nabla f(\mathbf{x}; \xi) - \nabla f(\mathbf{y}; \xi)\| \leq L\|\mathbf{x} - \mathbf{y}\|, \forall \mathbf{x}, \mathbf{y}$.*

---
**Algorithm 3** SPGR with Increasing Mini-Batch sizes: SPGR-imb
---
1: **Initialize**: $\mathbf{x}_0 \in \mathbb{R}^d$, $\eta = \frac{c}{L}$ with $0 < c < \frac{1}{6}$, $b \geq 1$
2: **Set**: $t = 0$, $\mathbf{x}_{-1} = \mathbf{x}_0$
3: **for** $s = 1, \ldots, S$ **do**
4:     Draw samples $\mathcal{S}_{1,s}$, let $\mathbf{g}_t = \nabla f_{\mathcal{S}_{1,s}}(\mathbf{x}_t)$                         $\diamond |\mathcal{S}_{1,s}| = b^2 s^2$
5:     $\mathbf{x}_{t+1} \in \text{prox}_{\eta r}[\mathbf{x}_t - \eta \mathbf{g}_t]$, $t = t + 1$
6:     **for** $q = 1, \ldots, bs$ **do**
7:        Draw samples $\mathcal{S}_{2,s}$, let $\mathbf{g}_t = \nabla f_{\mathcal{S}_{2,s}}(\mathbf{x}_t) - \nabla f_{\mathcal{S}_{2,s}}(\mathbf{x}_{t-1}) + \mathbf{g}_{t-1}$      $\diamond |\mathcal{S}_{2,s}| = bs$
8:        $\mathbf{x}_{t+1} \in \text{prox}_{\eta r}[\mathbf{x}_t - \eta \mathbf{g}_t]$, $t = t + 1$
9:     **end for**
10: **end for**
11: **Output**: $\mathbf{x}_R$, where $R$ is uniformly sampled from $\{1, \ldots, T\}$.
---

First, we present a general non-asymptotic convergence result of SPGR, which is summarized below.

**Theorem 5.** *Suppose Assumptions 1 and 2 hold, run Algorithm 2 with $\eta = \frac{c}{L}$ ($0 < c < \frac{1}{3}$) and $q = |\mathcal{S}_2|$, then the output $\mathbf{x}_R$ of Algorithm 2 satisfies* $\mathrm{E}[dist(0, \hat{\partial}F(\mathbf{x}_R))^2] \leq \frac{2\theta\Delta + \gamma\eta\Delta}{\eta\theta T} + \frac{(\gamma + 4\theta L)\sigma^2}{2\theta L |\mathcal{S}_1|}$ *for* **online setting** *and* $\mathrm{E}[dist(0, \hat{\partial}F(\mathbf{x}_R))^2] \leq \frac{2\theta\Delta + \gamma\eta\Delta}{\eta\theta T}$ *for* **finite-sum setting***, where $\gamma = 4L^2 + \frac{1}{\eta^2} + \frac{2L}{\eta}$ and $\theta = \frac{1-3\eta L}{2\eta}$ are two positive constants.*

Although the SARAH/SPIDER update used in Algorithm 2 is similar to that used in [41, 35] for handling convex regularizers, our analysis has some key differences from that in [41, 35]. In particular, the analysis in [41, 35] heavily relies on the convexity of the regularizer. In addition, they proved the convergence of the proximal gradient defined as $\mathcal{G}_\eta(\mathbf{x}) = \frac{1}{\eta}(\mathbf{x} - \text{prox}_{\eta r}(\mathbf{x} - \eta \nabla f(\mathbf{x})))$, while we directly prove the convergence of the subgradient $\hat{\partial}F(\mathbf{x})$. The convergence of the proximal gradient only implies a weak convergence of subgradient (i.e., a solution $\mathbf{x}$ which satisfies $\|\mathcal{G}_\eta(\mathbf{x})\| \leq \epsilon$ indicates that it is close to a solution $\mathbf{x}^+ = \text{prox}_{\eta r}(\mathbf{x} - \eta \nabla f(\mathbf{x}))$ such that $\|\hat{\partial}F(\mathbf{x}^+)\| \leq O(\epsilon)$ when $\eta = \Theta(1/L)$). The following corollary summarizes results in the two settings.

**Corollary 6.** *Under the same conditions and notations as in Theorem 5, in order to have* $\mathrm{E}[dist(0, \hat{\partial}F(\mathbf{x}_R))] \leq \epsilon$ *we can set:*

- *(**Online setting**) $q = |\mathcal{S}_2| = \sqrt{|\mathcal{S}_1|}$, $|\mathcal{S}_1| = \frac{(\gamma + 4\theta L)\sigma^2}{\theta L \epsilon^2}$, and $T = \frac{2(2\theta + \gamma\eta)\Delta}{\eta\theta\epsilon^2}$, giving a total complexity of $O(\epsilon^{-3})$.*
- *(**Finite-sum setting**) $q = |\mathcal{S}_2| = \sqrt{n}$, $|\mathcal{S}_1| = n$, and $T = \frac{(2\theta + \gamma\eta)\Delta}{\eta\theta\epsilon^2}$, leading to a total complexity of $O(\sqrt{n}\epsilon^{-2} + n)$.*

**Remark:** It is notable that the above complexity result is near-optimal according to [18, 50] for the finite-sum setting. For same special cases of $r(\mathbf{x})$, similar complexities have been established when $r(\mathbf{x}) = 0$ [18, 51] or when $r(\mathbf{x})$ is convex [41, 35].

## 4.1 SPGR with Increasing Mini-Batch Sizes

One limitation of SPGR for the online setting is that it requires knowing the target accuracy level $\epsilon$ in order to set $q$ and the sizes of $\mathcal{S}_1$ and $\mathcal{S}_2$, which makes it not practical. An user will need to worry about what is the right value of $\epsilon$ for running the algorithm, as a small $\epsilon$ may waste at lot of computations and a relatively large $\epsilon$ may not lead to an accurate solution. To address this issue, we propose a practical variant of SPGR, namely SPGR-imb, which uses increasing mini-batch sizes. The detailed updates are presented in Algorithm 3. The key idea is that we divide the whole progress into $S$ stages, and for each stage $s \in [S]$, the mini-batch sizes $|\mathcal{S}_1|$ and $|\mathcal{S}_2|$ are set to be proportional $s^2$ and $s$, respectively. The insight of this design is similar to Algorithm 1 with increasing mini-batch sizes, i.e., at earlier stages when the solution is far from a stationary solution we can tolerate a large variance in the stochastic gradient estimator and hence allow for a smaller mini-batch size. We summarize the non-asymptotic convergence result of SPGR-imb in the following theorem.

**Theorem 7.** *Suppose Assumptions 1 and 2 hold, run Algorithm 3 with $\eta = \frac{c}{L}$ ($0 < c < \frac{1}{3}$) and $S$ satisfying $bS(S+1)/2 = T$, then the output $\mathbf{x}_R$ of Algorithm 3 satisfies* $\mathrm{E}[dist(0, \hat{\partial}F(\mathbf{x}_R))^2] \leq \frac{(2\theta + \gamma\eta)\Delta}{\theta\eta T} + \frac{(4\theta L + \gamma)\sigma^2(\log(2T/b)+2)}{4b\theta LT}$ *for* **online setting** *and* $\mathrm{E}[dist(0, \hat{\partial}F(\mathbf{x}_R))^2] \leq \frac{(2\theta + \gamma\eta)\Delta}{\theta\eta T}$ *for* **finite-sum setting***, where $\gamma = 4L^2 + \frac{1}{\eta^2} + \frac{2L}{\eta}$ and $\theta = \frac{1-3\eta L}{2\eta}$ are two positive constants. In*

*particular in order to have* $\mathrm{E}[dist(0, \hat{\partial} F(\mathbf{x}_R))] \leq \epsilon$, *it suffices to set* $T = \widetilde{O}(1/\epsilon^2)$. *The total complexity is* $\widetilde{O}(1/\epsilon^3)$.

**Remark:** Compared to the result in Corollary 6, the complexity result of Theorem 7 is only worse by a logarithmic factor.

## 5   Experiments

**Regularized loss minimization.** First, we compare MB-SPG, SPGR with MBSGA, VRSGA, SSDC-SPG and SSDC-SVRG for solving the regularized non-linear least square (NLLS) classification problems $\frac{1}{n} \sum_{i=1}^n (b_i - \sigma(\mathbf{x}^\top \mathbf{a}_i))^2 + r(\mathbf{x})$ with a sigmod function $\sigma(s) = \frac{1}{1+e^{-s}}$ for classification, and the regularized truncated least square (TLS) loss function $\frac{1}{2n} \sum_{i=1}^n \alpha \log(1 + (y_i - \mathbf{w}^\top \mathbf{x}_i)^2 / \alpha) + r(\mathbf{x})$ for regression [44]. Two data sets (covtype and a9a) are used for classification, and two data sets E2006 and triazines are used for regression. These data sets are downloaded from the libsvm website. We use three different non-smooth non-convex regularizers, i.e., $\ell_0$ regularizer $r(\mathbf{x}) = \lambda \|\mathbf{x}\|_0$, $\ell_{0.5}$ regularizer $r(\mathbf{x}) = \lambda \|\mathbf{x}\|_{0.5}$, and indicator function of $\ell_0$ constraint $I_{\{\|\mathbf{x}\|_0 \leq \kappa\}}(\mathbf{x})$. The truncation value $\alpha$ is set to $\sqrt{10n}$ following [44]. The value of regularization parameter $\lambda$ is fixed as $10^{-4}$ and the value of $\kappa$ is fixed as $0.2d$ where $d$ is the dimension of data. For all algorithms, we use the theoretical values of the parameters for the sake of fairness in comparison. All algorithms start with the same initial solution with all zero entries. We implement the increasing mini-batch versions of MB-SPG and SPGR (online setting) with $b = 1$. The unknown parameter $\sigma$ in MBSGA is estimated following [29]. The objective value (in log scale) versus the number of gradient computations for different tasks are plotted in Figure 1. The solid lines correspond to algorithms running in the online setting and the dashed lines correspond to algorithms running in the finite-sum setting. By comparing algorithms running in the online setting including MB-SPG, SPGR, MBSGA and SSDC-SPG, we can see that the proposed algorithms (MB-SPG and SPGR) are faster across different tasks. In addition, SPGR is faster than MB-SPG. These results are consistent with our theory. By comparing algorithms running in the finite-sum setting including VRSGA, SSDC-SVRG and SPGR, we can see that the proposed SPGR is much faster, which also corroborates our theory.

**Learning with Quantization**. Second, we consider the problem of learning a quantized model where the model parameter is represented by a small number of bits (e.g., 2 bits that can encode 1 or −1). It has received tremendous attention in deep learning for model compression [21, 42, 36]. An idea to formulate the problem is to consider a constrained optimization problem: $\min_{\mathbf{x} \in \Omega} f(\mathbf{x})$ where $\Omega$ denotes a discrete set including the values that can be represented by a small number of bits. However, finding a stationary point for this problem is meaningless. This is because that for a discrete set $\Omega$, the subgradient of its indicator function $I_\Omega(\mathbf{x})$ is the whole space [13, 22]. Hence, we have $0 \in \hat{\partial}(f(\mathbf{x}) + I_\Omega(\mathbf{x}))$ for any $\mathbf{x} \in \Omega$. To avoid this issue, we consider a different formulation by using a penalization of the constraint: $\min_{\mathbf{x} \in \mathbb{R}^d} f(\mathbf{x}) + \frac{\lambda}{2} \|\mathbf{x} - P_\Omega(\mathbf{x})\|^2$, where $P_\Omega(\mathbf{x})$ is a projection onto the set $\Omega$ and $\lambda > 0$ is a penalization parameter. This penalization-based approach is one standard way to handle complicated constraints [4, 28]. It is notable that in general the penalization term is a non-smooth non-convex function of $\mathbf{x}$ for a non-convex set $\Omega$, though its local smoothness has been proved under some regularity condition of $\Omega$ [37]. The proximal mapping of the penalization term has a closed-form solution as long as $P_\Omega(\mathbf{x})$ can be easily computed [24], which corresponds to quantization for our considered problem.

In the experiment, we use the NLLS loss similar to regularized loss minimization for learning a quantized non-linear model, and focus on comparison of algorithms running in the online setting including MBSGA, SSDC-SPG, MB-SPG and SPGR. We also implement a popular heuristic SGD approach in deep learning for learning a quantized model [36], which updates the solution simply by $\mathbf{x}_{t+1} = \mathbf{x}_t - \eta_t \nabla f(\hat{\mathbf{x}}_t; \xi_t)$ where $\hat{\mathbf{x}}_t = P_\Omega(\mathbf{x}_t)$ is the quantized model. We conduct the experiments on four data sets mnist, news20, rcv1, w8a, where the last three data sets are downloaded from the libsvm website. We compare the testing accuracy of learned quantized model versus the number of iterations, and the results are plotted in Figure 2, where $q$ denotes the number of bits for quantization. We fix $\lambda = 1$, and decrease the step size by half every 100 iterations for heuristic SGD, MBSGA and MB-SPG. This is helpful for generalization purpose. We can see that the proposed SPGR algorithm has better testing accuracy in most cases, and the proposed MB-SPG has comparable performance if not better results than other baselines.

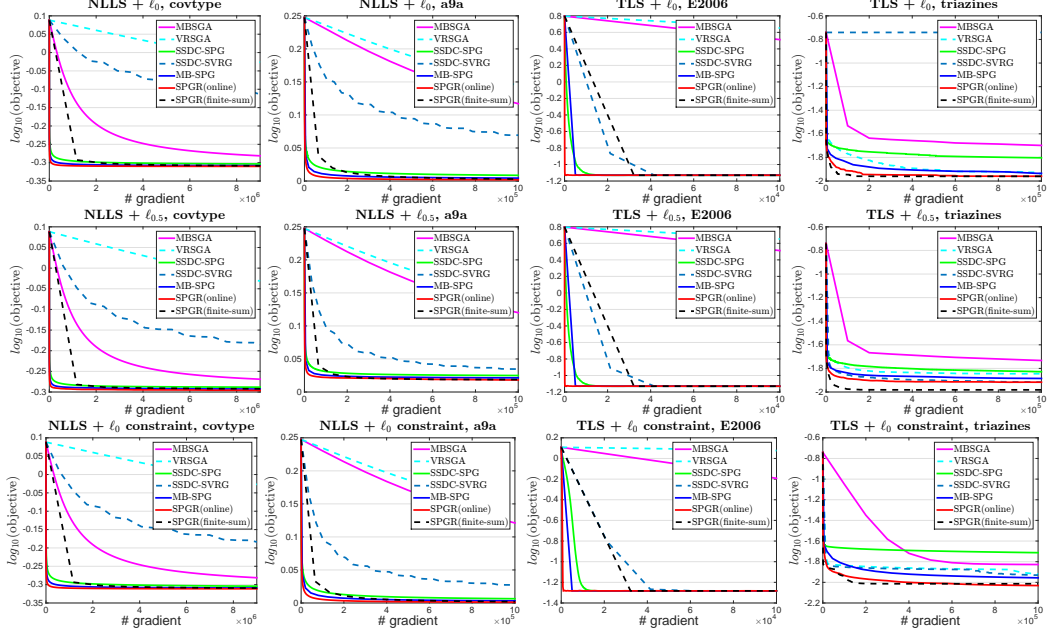

Figure 1: Comparisons of different algorithms for regularized loss minimization.

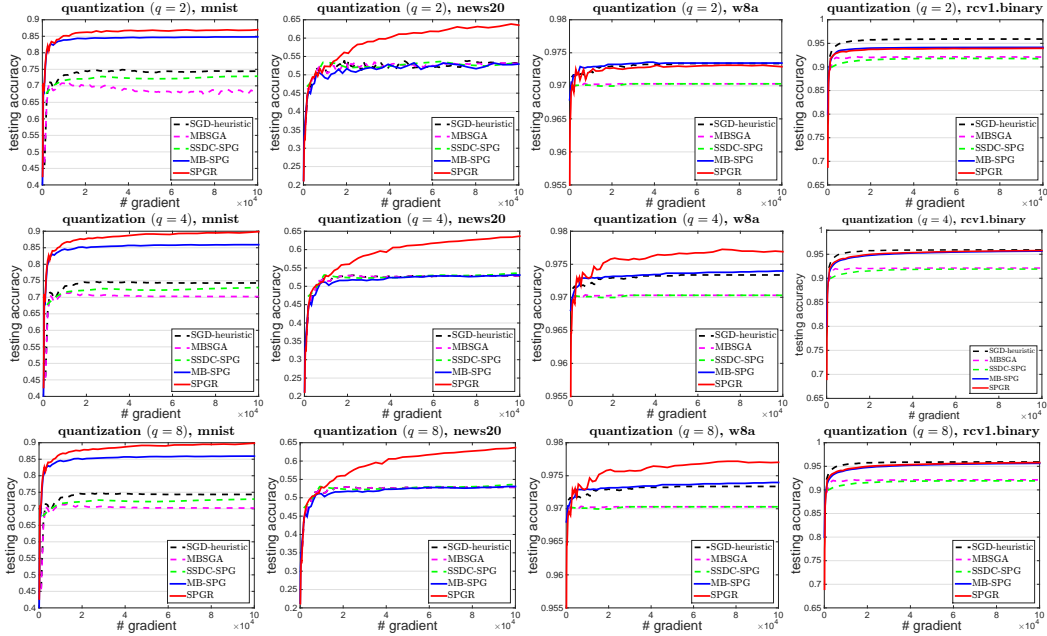

Figure 2: Comparisons of different algorithms for learning with quantization.

# 6 Conclusions

In this paper, we have presented the first non-asymptotic convergence analysis of stochastic proximal gradient methods for solving a non-convex optimization problem with a smooth loss function and a non-smooth non-convex regularizer. The proposed algorithms enjoy improved complexities than the state-of-the-art results for the same problems, and also match the existing complexity results for solving non-convex minimization problems with a smooth loss and a non-smooth convex regularizer.

## Acknowledgements

The authors thank the anonymous reviewers for their helpful comments. Y. Xu and T. Yang are partially supported by National Science Foundation (IIS-1545995).

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
