[Supplementary Material]

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

# A  Proof of Theorem 1

Before starting the proof, we give the detailed updates of PGD in Algorithm 4.

---

**Algorithm 4** Proximal Gradient Descent: PGD($\mathbf{x}_0$, $T$, $L$, $c$)

---
1: **Input:** $\mathbf{x}_0 \in \mathbb{R}^d$, the number of iterations $T$, $\eta = \frac{c}{L}$ with $0 < c < 1$.
2: **for** $t = 0, 1, \ldots, T - 1$ **do**
3:    $\mathbf{x}_{t+1} \in \text{prox}_{\eta r}[\mathbf{x}_t - \eta \nabla f(\mathbf{x}_t)]$
4: **end for**
5: **Output:** $\mathbf{x}_R$, where $R$ is uniformly sampled from $\{1, \ldots, T\}$.

---

*Proof.* Based on the update of Algorithm 4, by Exercise 8.8 and Theorem 10.1 of [39] we know

$$-\nabla f(\mathbf{x}_t) - \frac{1}{\eta}(\mathbf{x}_{t+1} - \mathbf{x}_t) \in \hat{\partial} r(\mathbf{x}_{t+1}),$$

which implies that

$$\nabla f(\mathbf{x}_{t+1}) - \nabla f(\mathbf{x}_t) - \frac{1}{\eta}(\mathbf{x}_{t+1} - \mathbf{x}_t) \in \nabla f(\mathbf{x}_{t+1}) + \hat{\partial} r(\mathbf{x}_{t+1}) = \hat{\partial} F(\mathbf{x}_{t+1}). \tag{6}$$

By the update of (5), we also have

$$r(\mathbf{x}_{t+1}) + \langle \nabla f(\mathbf{x}_t), \mathbf{x}_{t+1} - \mathbf{x}_t \rangle + \frac{1}{2\eta}\|\mathbf{x}_{t+1} - \mathbf{x}_t\|^2 \le r(\mathbf{x}_t). \tag{7}$$

Since $f(\mathbf{x})$ is smooth with parameter $L$, then

$$f(\mathbf{x}_{t+1}) \le f(\mathbf{x}_t) + \langle \nabla f(\mathbf{x}_t), \mathbf{x}_{t+1} - \mathbf{x}_t \rangle + \frac{L}{2}\|\mathbf{x}_{t+1} - \mathbf{x}_t\|^2. \tag{8}$$

Combining these two inequalities (7) and (8) and using the fact that $F(\mathbf{x}) = f(\mathbf{x}) + r(\mathbf{x})$, we get

$$\frac{1}{2}(1/\eta - L)\|\mathbf{x}_{t+1} - \mathbf{x}_t\|^2 \le F(\mathbf{x}_t) - F(\mathbf{x}_{t+1}). \tag{9}$$

By summing the above inequalities across $t = 0, \ldots, T - 1$ and using $F(\mathbf{x}_*) \le F(\mathbf{x})$ for any $\mathbf{x} \in \mathbb{R}^d$ and the Assumption 1 (iii), we know

$$\frac{1}{2}(1/\eta - L)\sum_{t=0}^{T-1} \|\mathbf{x}_{t+1} - \mathbf{x}_t\|^2 \le F(\mathbf{x}_0) - F(\mathbf{x}_T) \le F(\mathbf{x}_0) - F(\mathbf{x}_*) \le \Delta. \tag{10}$$

On the other hand, by Young's inequality $\|\mathbf{a} \pm \mathbf{b}\|^2 \le 2\|\mathbf{a}\|^2 + 2\|\mathbf{b}\|^2$ and the smoothness of $f(\mathbf{x})$,

$$\|\nabla f(\mathbf{x}_{t+1}) - \nabla f(\mathbf{x}_t) - \frac{1}{\eta}(\mathbf{x}_{t+1} - \mathbf{x}_t)\|^2$$

$$\le 2\|\nabla f(\mathbf{x}_{t+1}) - \nabla f(\mathbf{x}_t)\|^2 + \frac{2}{\eta^2}\|\mathbf{x}_{t+1} - \mathbf{x}_t\|^2$$

$$\le 2(L^2 + \frac{1}{\eta^2})\|\mathbf{x}_{t+1} - \mathbf{x}_t\|^2.$$

Therefore, summing the above inequalities across $t = 0, \ldots, T - 1$ and using the inequality (10) with $1/\eta - L > 0$,

$$\frac{1}{T}\sum_{t=0}^{T-1} \|\nabla f(\mathbf{x}_{t+1}) - \nabla f(\mathbf{x}_t) - \frac{1}{\eta}(\mathbf{x}_{t+1} - \mathbf{x}_t)\|^2$$

$$\le 2(L^2 + \frac{1}{\eta^2})\frac{1}{T}\sum_{t=0}^{T-1} \|\mathbf{x}_{t+1} - \mathbf{x}_t\|^2$$

$$\le \frac{2(L^2 + \frac{1}{\eta^2})}{\frac{1}{2}(1/\eta - L)T}\Delta = \frac{4(\eta^2 L^2 + 1)}{\eta(1 - \eta L)T}\Delta.$$

By (6) we know

$$\text{dist}(0, \hat{\partial} F(\mathbf{x}_{t+1}))^2 \le \|\nabla f(\mathbf{x}_{t+1}) - \nabla f(\mathbf{x}_t) - \frac{1}{\eta}(\mathbf{x}_{t+1} - \mathbf{x}_t)\|^2,$$

then by the fact that $R$ is uniformly sampled from $\{1, \ldots, T\}$,

$$\mathrm{E}[\mathrm{dist}(0, \hat{\partial} F(\mathbf{x}_R))^2] = \frac{1}{T} \sum_{t=0}^{T-1} \mathrm{dist}(0, \hat{\partial} F(\mathbf{x}_{t+1}))^2 \leq \frac{4(\eta^2 L^2 + 1)}{\eta(1 - \eta L)T} \Delta.$$

By the setting of $\eta = \frac{c}{L} < \frac{1}{L}$, and let $T = \frac{4(\eta^2 L^2 + 1)}{\eta(1 - \eta L)\epsilon^2} \Delta = O(1/\epsilon^2)$, we get

$$\mathrm{E}[\mathrm{dist}(0, \hat{\partial} F(\mathbf{x}_R))^2] \leq \epsilon^2.$$

By using the fact that $(\mathrm{E}[\mathrm{dist}(0, \hat{\partial} F(\mathbf{x}_R))])^2 \leq \mathrm{E}[\mathrm{dist}(0, \hat{\partial} F(\mathbf{x}_R))^2]$, we have

$$\mathrm{E}[\mathrm{dist}(0, \hat{\partial} F(\mathbf{x}_R))] \leq \epsilon.$$

$\square$

## B  Proof of Theorem 2

*Proof.* Recall that the update of $\mathbf{x}_{t+1}$ is

$$\mathbf{x}_{t+1} \in \arg\min_{\mathbf{x} \in \mathbb{R}^d} \left\{ r(\mathbf{x}) + \frac{1}{2\eta} \|\mathbf{x} - (\mathbf{x}_t - \eta \mathbf{g}_t)\|^2 \right\}$$

$$= \arg\min_{\mathbf{x} \in \mathbb{R}^d} \left\{ r(\mathbf{x}) + \langle \mathbf{g}_t, \mathbf{x} - \mathbf{x}_t \rangle + \frac{1}{2\eta} \|\mathbf{x} - \mathbf{x}_t\|^2 \right\},$$

then by Exercise 8.8 and Theorem 10.1 of [39] we know

$$-\mathbf{g}_t - \frac{1}{\eta}(\mathbf{x}_{t+1} - \mathbf{x}_t) \in \hat{\partial} r(\mathbf{x}_{t+1}),$$

which implies that

$$\nabla f(\mathbf{x}_{t+1}) - \mathbf{g}_t - \frac{1}{\eta}(\mathbf{x}_{t+1} - \mathbf{x}_t) \in \nabla f(\mathbf{x}_{t+1}) + \hat{\partial} r(\mathbf{x}_{t+1}) = \hat{\partial} F(\mathbf{x}_{t+1}). \tag{11}$$

By the update of $\mathbf{x}_{t+1}$ in Algorithm 1, we also have

$$r(\mathbf{x}_{t+1}) + \langle \mathbf{g}_t, \mathbf{x}_{t+1} - \mathbf{x}_t \rangle + \frac{1}{2\eta} \|\mathbf{x}_{t+1} - \mathbf{x}_t\|^2 \leq r(\mathbf{x}_t). \tag{12}$$

Since $f(\mathbf{x})$ is smooth with parameter $L$, then

$$f(\mathbf{x}_{t+1}) \leq f(\mathbf{x}_t) + \langle \nabla f(\mathbf{x}_t), \mathbf{x}_{t+1} - \mathbf{x}_t \rangle + \frac{L}{2} \|\mathbf{x}_{t+1} - \mathbf{x}_t\|^2. \tag{13}$$

Combining these two inequalities (12) and (13) we get

$$\langle \mathbf{g}_t - \nabla f(\mathbf{x}_t), \mathbf{x}_{t+1} - \mathbf{x}_t \rangle + \frac{1}{2}(1/\eta - L)\|\mathbf{x}_{t+1} - \mathbf{x}_t\|^2 \leq F(\mathbf{x}_t) - F(\mathbf{x}_{t+1}). \tag{14}$$

That is

$$\frac{1}{2}(1/\eta - L)\|\mathbf{x}_{t+1} - \mathbf{x}_t\|^2 \leq F(\mathbf{x}_t) - F(\mathbf{x}_{t+1}) - \langle \mathbf{g}_t - \nabla f(\mathbf{x}_t), \mathbf{x}_{t+1} - \mathbf{x}_t \rangle$$

$$\leq F(\mathbf{x}_t) - F(\mathbf{x}_{t+1}) + \frac{1}{2L} \|\mathbf{g}_t - \nabla f(\mathbf{x}_t)\|^2 + \frac{L}{2} \|\mathbf{x}_{t+1} - \mathbf{x}_t\|^2,$$

where the last inequality uses Young's inequality $\langle \mathbf{a}, \mathbf{b} \rangle \leq \frac{1}{2}\|\mathbf{a}\|^2 + \frac{1}{2}\|\mathbf{b}\|^2$. Then by rearranging above inequality and summing it across $t = 0, \ldots, T - 1$, we have

$$\frac{1 - 2\eta L}{2\eta} \sum_{t=0}^{T-1} \|\mathbf{x}_{t+1} - \mathbf{x}_t\|^2 \leq F(\mathbf{x}_0) - F(\mathbf{x}_T) + \frac{1}{2L} \sum_{t=0}^{T-1} \|\mathbf{g}_t - \nabla f(\mathbf{x}_t)\|^2$$

$$\leq F(\mathbf{x}_0) - F(\mathbf{x}_*) + \frac{1}{2L} \sum_{t=0}^{T-1} \|\mathbf{g}_t - \nabla f(\mathbf{x}_t)\|^2$$

$$\leq \Delta + \frac{1}{2L} \sum_{t=0}^{T-1} \|\mathbf{g}_t - \nabla f(\mathbf{x}_t)\|^2, \tag{15}$$

where the second inequality uses the fact that $F(\mathbf{x}_*) \leq F(\mathbf{x})$ for any $\mathbf{x} \in \mathbb{R}^d$ and the last inequality uses the Assumption 1 (iii).

On the other hand, by (14) we get

$$\langle \mathbf{g}_t - \nabla f(\mathbf{x}_{t+1}), \mathbf{x}_{t+1} - \mathbf{x}_t \rangle + \frac{1}{2}(1/\eta - L)\|\mathbf{x}_{t+1} - \mathbf{x}_t\|^2$$
$$\leq F(\mathbf{x}_t) - F(\mathbf{x}_{t+1}) - \langle \nabla f(\mathbf{x}_{t+1}) - \nabla f(\mathbf{x}_t), \mathbf{x}_{t+1} - \mathbf{x}_t \rangle$$

i.e.,

$$\frac{2}{\eta}\langle \mathbf{g}_t - \nabla f(\mathbf{x}_{t+1}), \mathbf{x}_{t+1} - \mathbf{x}_t \rangle + \frac{1 - \eta L}{\eta^2}\|\mathbf{x}_{t+1} - \mathbf{x}_t\|^2$$
$$\leq \frac{2(F(\mathbf{x}_t) - F(\mathbf{x}_{t+1}))}{\eta} - \frac{2}{\eta}\langle \nabla f(\mathbf{x}_{t+1}) - \nabla f(\mathbf{x}_t), \mathbf{x}_{t+1} - \mathbf{x}_t \rangle. \tag{16}$$

Since $2\langle \mathbf{g}_t - \nabla f(\mathbf{x}_{t+1}), \frac{1}{\eta}(\mathbf{x}_{t+1} - \mathbf{x}_t)\rangle = \|\mathbf{g}_t - \nabla f(\mathbf{x}_{t+1}) + \frac{1}{\eta}(\mathbf{x}_{t+1} - \mathbf{x}_t)\|^2 - \|\mathbf{g}_t - \nabla f(\mathbf{x}_{t+1})\|^2 - \frac{1}{\eta^2}\|\mathbf{x}_{t+1} - \mathbf{x}_t\|^2$, then plugging above inequality into (16) and rearranging it we have

$$\|\mathbf{g}_t - \nabla f(\mathbf{x}_{t+1}) + \frac{1}{\eta}(\mathbf{x}_{t+1} - \mathbf{x}_t)\|^2$$
$$\leq \|\mathbf{g}_t - \nabla f(\mathbf{x}_{t+1})\|^2 + \frac{1}{\eta^2}\|\mathbf{x}_{t+1} - \mathbf{x}_t\|^2 - \frac{1 - \eta L}{\eta^2}\|\mathbf{x}_{t+1} - \mathbf{x}_t\|^2$$
$$+ \frac{2(F(\mathbf{x}_t) - F(\mathbf{x}_{t+1}))}{\eta} - \frac{2}{\eta}\langle \nabla f(\mathbf{x}_{t+1}) - \nabla f(\mathbf{x}_t), \mathbf{x}_{t+1} - \mathbf{x}_t \rangle$$
$$\leq 2\|\mathbf{g}_t - \nabla f(\mathbf{x}_t)\|^2 + 2\|\nabla f(\mathbf{x}_t) - \nabla f(\mathbf{x}_{t+1})\|^2 + \frac{1}{\eta^2}\|\mathbf{x}_{t+1} - \mathbf{x}_t\|^2$$
$$- \frac{1 - \eta L}{\eta^2}\|\mathbf{x}_{t+1} - \mathbf{x}_t\|^2 + \frac{2(F(\mathbf{x}_t) - F(\mathbf{x}_{t+1}))}{\eta} - \frac{2}{\eta}\langle \nabla f(\mathbf{x}_{t+1}) - \nabla f(\mathbf{x}_t), \mathbf{x}_{t+1} - \mathbf{x}_t \rangle$$
$$\leq 2\|\mathbf{g}_t - \nabla f(\mathbf{x}_t)\|^2 + 2L^2\|\mathbf{x}_t - \mathbf{x}_{t+1}\|^2 + \frac{1}{\eta^2}\|\mathbf{x}_{t+1} - \mathbf{x}_t\|^2$$
$$- \frac{1 - \eta L}{\eta^2}\|\mathbf{x}_{t+1} - \mathbf{x}_t\|^2 + \frac{2(F(\mathbf{x}_t) - F(\mathbf{x}_{t+1}))}{\eta} + \frac{2L}{\eta}\|\mathbf{x}_{t+1} - \mathbf{x}_t\|^2$$
$$= 2\|\mathbf{g}_t - \nabla f(\mathbf{x}_t)\|^2 + \frac{2(F(\mathbf{x}_t) - F(\mathbf{x}_{t+1}))}{\eta} + (2L^2 + \frac{3L}{\eta})\|\mathbf{x}_{t+1} - \mathbf{x}_t\|^2,$$

where the second inequality is due to Young's inequality $\|\mathbf{a} \pm \mathbf{b}\|^2 \leq 2\|\mathbf{a}\|^2 + 2\|\mathbf{b}\|^2$; the last inequality is due to the Assumption 1 (iv) of $\|\nabla f(\mathbf{x}) - \nabla f(\mathbf{y})\| \leq L\|\mathbf{x} - \mathbf{y}\|$ for any $\mathbf{x}, \mathbf{y} \in \mathbb{R}^d$ and Cauchy-Schwartz inequality. By summing up $t = 0, 1, \ldots, T - 1$, we have

$$\sum_{t=0}^{T-1} \|\mathbf{g}_t - \nabla f(\mathbf{x}_{t+1}) + \frac{1}{\eta}(\mathbf{x}_{t+1} - \mathbf{x}_t)\|^2$$
$$\leq 2\sum_{t=0}^{T-1} \|\mathbf{g}_t - \nabla f(\mathbf{x}_t)\|^2 + \frac{2(F(\mathbf{x}_0) - F(\mathbf{x}_T))}{\eta} + (2L^2 + \frac{3L}{\eta})\sum_{t=0}^{T-1} \|\mathbf{x}_t - \mathbf{x}_{t+1}\|^2$$
$$\leq 2\sum_{t=0}^{T-1} \|\mathbf{g}_t - \nabla f(\mathbf{x}_t)\|^2 + \frac{2(F(\mathbf{x}_0) - F(\mathbf{x}_*))}{\eta} + (2L^2 + \frac{3L}{\eta})\sum_{t=0}^{T-1} \|\mathbf{x}_t - \mathbf{x}_{t+1}\|^2$$
$$\leq 2\sum_{t=0}^{T-1} \|\mathbf{g}_t - \nabla f(\mathbf{x}_t)\|^2 + \frac{2\Delta}{\eta} + \frac{2}{\eta^2}\sum_{t=0}^{T-1} \|\mathbf{x}_t - \mathbf{x}_{t+1}\|^2,$$

where the second inequality is due to $F(\mathbf{x}_*) \leq F(\mathbf{x}_T)$; the last inequality holds by setting $\eta = \frac{c}{L} < \frac{1}{2L}$ and Assumption 1(iii) of $F(\mathbf{x}_0) - F(\mathbf{x}_*) \leq \Delta$. Combining above inequality with (11) and (15)

and taking the expectation, we have

$$\mathrm{E}_R[\mathrm{dist}(0, \hat{\partial} F(\mathbf{x}_R))^2]$$

$$\leq \frac{1}{T} \sum_{t=0}^{T-1} \mathrm{E}[\|\mathbf{g}_t - \nabla f(\mathbf{x}_{t+1}) + \frac{1}{\eta}(\mathbf{x}_{t+1} - \mathbf{x}_t)\|^2]$$

$$\leq \frac{2}{T} \sum_{t=0}^{T-1} \mathrm{E}[\|\mathbf{g}_t - \nabla f(\mathbf{x}_t)\|^2] + \frac{2\Delta}{\eta T} + \frac{2}{\eta^2 T}\left(\frac{2}{1/\eta - 2L}\Delta + \frac{1}{L/\eta - 2L^2}\sum_{t=0}^{T-1} \mathrm{E}[\|\mathbf{g}_t - \nabla f(\mathbf{x}_t)\|^2]\right)$$

$$= \frac{2c(1-2c)+2}{c(1-2c)}\frac{1}{T}\sum_{t=0}^{T-1}\mathrm{E}[\|\mathbf{g}_t - \nabla f(\mathbf{x}_t)\|^2] + \frac{6-4c}{1-2c}\frac{\Delta}{\eta T},$$

where $0 < c < \frac{1}{2}$. $\qquad\square$

## C  Proof of Theorem 5

Before starting the proof, we present the error bound of the SARAH/SPIDER estimator in the following lemma from [18] that will be used in the proof.

**Lemma 1** (Lemma 1 [18]). *Suppose that Assumptions 1 and 2 hold, then for any $t$ such that $(n_t - 1)q \leq t \leq n_t q - 1$ with $n_t = \lceil t/q \rceil$ in Algorithm 2, we have*

$$\mathrm{E}[\|\mathbf{g}_t - \nabla f(\mathbf{x}_t)\|^2] \leq \frac{L^2}{|\mathcal{S}_2|}\sum_{i=(n_t-1)q}^{t} \mathrm{E}[\|\mathbf{x}_{i+1} - \mathbf{x}_i\|^2] + \mathrm{E}[\|\mathbf{g}_{(n_t-1)q} - \nabla f(\mathbf{x}_{(n_t-1)q})\|^2].$$

*Proof of Theorem 5.* We first focus on the online setting. Similar to the proof of Theorem 2 we have

$$\nabla f(\mathbf{x}_{t+1}) - \mathbf{g}_t - \frac{1}{\eta}(\mathbf{x}_{t+1} - \mathbf{x}_t) \in \nabla f(\mathbf{x}_{t+1}) + \hat{\partial} r(\mathbf{x}_{t+1}) = \hat{\partial} F(\mathbf{x}_{t+1}). \tag{17}$$

And we also have

$$F(\mathbf{x}_{t+1}) - F(\mathbf{x}_t) \leq -\langle \mathbf{g}_t - \nabla f(\mathbf{x}_t), \mathbf{x}_{t+1} - \mathbf{x}_t \rangle - \frac{1}{2}(1/\eta - L)\|\mathbf{x}_{t+1} - \mathbf{x}_t\|^2$$

$$\leq \frac{1}{2L}\|\mathbf{g}_t - \nabla f(\mathbf{x}_t)\|^2 + \frac{L}{2}\|\mathbf{x}_{t+1} - \mathbf{x}_t\|^2 - \frac{1}{2}(1/\eta - L)\|\mathbf{x}_{t+1} - \mathbf{x}_t\|^2$$

$$= \frac{1}{2L}\|\mathbf{g}_t - \nabla f(\mathbf{x}_t)\|^2 - \frac{1}{2}(1/\eta - 2L)\|\mathbf{x}_{t+1} - \mathbf{x}_t\|^2, \tag{18}$$

where the second inequality uses Young's inequality $\langle \mathbf{a}, \mathbf{b}\rangle \leq \frac{1}{2}\|\mathbf{a}\|^2 + \frac{1}{2}\|\mathbf{b}\|^2$. By taking the expectation on both sides of above inequality, we get

$$\mathrm{E}[F(\mathbf{x}_{t+1})] - \mathrm{E}[F(\mathbf{x}_t)] \leq \frac{1}{2L}\mathrm{E}[\|\mathbf{g}_t - \nabla f(\mathbf{x}_t)\|^2] - \frac{1 - 2\eta L}{2\eta}\mathrm{E}[\|\mathbf{x}_{t+1} - \mathbf{x}_t\|^2]. \tag{19}$$

Next, we want to upper bound the variance term $\mathrm{E}[\|\mathbf{g}_t - \nabla f(\mathbf{x}_t)\|^2]$ by using Lemma 1 of [18]. In particular, by Lemma 1, for any $t$ such that $(n_t - 1)q \leq t \leq n_t q - 1$ with $n_t = \lceil t/q \rceil$ in Algorithm 2, we have

$$\mathrm{E}[\|\mathbf{g}_t - \nabla f(\mathbf{x}_t)\|^2] \leq \frac{L^2}{|\mathcal{S}_2|}\sum_{i=(n_t-1)q}^{t}\mathrm{E}[\|\mathbf{x}_{i+1} - \mathbf{x}_i\|^2] + \mathrm{E}[\|\mathbf{g}_{(n_t-1)q} - \nabla f(\mathbf{x}_{(n_t-1)q})\|^2]. \tag{20}$$

Plugging inequality (20) into inequality (19),

$$\mathrm{E}[F(\mathbf{x}_{t+1})] - \mathrm{E}[F(\mathbf{x}_t)] \leq -\frac{1 - 2\eta L}{2\eta}\mathrm{E}[\|\mathbf{x}_{t+1} - \mathbf{x}_t\|^2]$$

$$+ \frac{1}{2L}\left(\frac{L^2}{|\mathcal{S}_2|}\sum_{i=(n_t-1)q}^{t}\mathrm{E}[\|\mathbf{x}_{i+1} - \mathbf{x}_i\|^2] + \mathrm{E}[\|\mathbf{g}_{(n_t-1)q} - \nabla f(\mathbf{x}_{(n_t-1)q})\|^2]\right). \tag{21}$$

By the updates of Algorithm 2, under Assumption 1 (ii) we have

$$\mathrm{E}[\|\mathbf{g}_{(n_t-1)q} - \nabla f(\mathbf{x}_{(n_t-1)q})\|^2] \leq \frac{\sigma^2}{|\mathcal{S}_1|}. \tag{22}$$

Then inequality (21) implies that

$$\mathrm{E}[F(\mathbf{x}_{t+1})] - \mathrm{E}[F(\mathbf{x}_t)]$$

$$\leq \frac{L}{2|\mathcal{S}_2|} \sum_{i=(n_t-1)q}^{t} \mathrm{E}[\|\mathbf{x}_{i+1} - \mathbf{x}_i\|^2] + \frac{\sigma^2}{2L|\mathcal{S}_1|} - \frac{1 - 2\eta L}{2\eta} \mathrm{E}[\|\mathbf{x}_{t+1} - \mathbf{x}_t\|^2]. \qquad (23)$$

For any $t$ such that $(n_t - 1)q \leq t \leq n_t q - 1$, we take the telescoping sum of (23) over $t$ from $(n_t - 1)q$ to $t$.

$$\mathrm{E}[F(\mathbf{x}_{t+1})] - \mathrm{E}[F(\mathbf{x}_{(n_t-1)q})]$$

$$\leq \frac{L}{2|\mathcal{S}_2|} \sum_{j=(n_t-1)q}^{t} \sum_{i=(n_t-1)q}^{j} \mathrm{E}[\|\mathbf{x}_{i+1} - \mathbf{x}_i\|^2] + \sum_{j=(n_t-1)q}^{t} \frac{\sigma^2}{2L|\mathcal{S}_1|} - \frac{1 - 2\eta L}{2\eta} \sum_{j=(n_t-1)q}^{t} \mathrm{E}[\|\mathbf{x}_{j+1} - \mathbf{x}_j\|^2]$$

$$\leq \frac{L}{2|\mathcal{S}_2|} \sum_{j=(n_t-1)q}^{t} \sum_{i=(n_t-1)q}^{t} \mathrm{E}[\|\mathbf{x}_{i+1} - \mathbf{x}_i\|^2] + \sum_{j=(n_t-1)q}^{t} \frac{\sigma^2}{2L|\mathcal{S}_1|} - \frac{1 - 2\eta L}{2\eta} \sum_{j=(n_t-1)q}^{t} \mathrm{E}[\|\mathbf{x}_{j+1} - \mathbf{x}_j\|^2]$$

$$\leq \frac{Lq}{2|\mathcal{S}_2|} \sum_{i=(n_t-1)q}^{t} \mathrm{E}[\|\mathbf{x}_{i+1} - \mathbf{x}_i\|^2] + \sum_{j=(n_t-1)q}^{t} \frac{\sigma^2}{2L|\mathcal{S}_1|} - \frac{1 - 2\eta L}{2\eta} \sum_{j=(n_t-1)q}^{t} \mathrm{E}[\|\mathbf{x}_{j+1} - \mathbf{x}_j\|^2]$$

$$= \sum_{j=(n_t-1)q}^{t} \frac{\sigma^2}{2L|\mathcal{S}_1|} - \theta \sum_{j=(n_t-1)q}^{t} \mathrm{E}[\|\mathbf{x}_{j+1} - \mathbf{x}_j\|^2],$$

where the second inequality is due to $j \leq t$; the third inequality is due to $(n_t - 1)q \leq t \leq n_t q - 1$; $\theta := \frac{1 - 2\eta L}{2\eta} - \frac{Lq}{2|\mathcal{S}_2|}$. Therefore we have

$$\mathrm{E}[F(\mathbf{x}_{t+1})] - \mathrm{E}[F(\mathbf{x}_{(n_t-1)q})] \leq \sum_{j=(n_t-1)q}^{t} \frac{\sigma^2}{2L|\mathcal{S}_1|} - \theta \sum_{j=(n_t-1)q}^{t} \mathrm{E}[\|\mathbf{x}_{j+1} - \mathbf{x}_j\|^2].$$

Then

$$\mathrm{E}[F(\mathbf{x}_T)] - \mathrm{E}[F(\mathbf{x}_0)]$$

$$= \mathrm{E}[F(\mathbf{x}_T)] - \mathrm{E}[F(\mathbf{x}_{(n_T-1)q})] + \cdots + \mathrm{E}[F(\mathbf{x}_{2q})] - \mathrm{E}[F(\mathbf{x}_q)] + \mathrm{E}[F(\mathbf{x}_q)] - \mathrm{E}[F(\mathbf{x}_0)]$$

$$\leq \sum_{j=0}^{T-1} \frac{\sigma^2}{2L|\mathcal{S}_1|} - \theta \sum_{j=0}^{T-1} \mathrm{E}[\|\mathbf{x}_{j+1} - \mathbf{x}_j\|^2]$$

$$= \frac{\sigma^2 T}{2L|\mathcal{S}_1|} - \theta \sum_{t=0}^{T-1} \mathrm{E}[\|\mathbf{x}_{t+1} - \mathbf{x}_t\|^2].$$

By the setting of $\eta$ such that $\theta > 0$, therefore above inequality becomes

$$\frac{1}{T} \sum_{t=0}^{T-1} \mathrm{E}[\|\mathbf{x}_{t+1} - \mathbf{x}_t\|^2] \leq \frac{\sigma^2}{2\theta L|\mathcal{S}_1|} + \frac{\mathrm{E}[F(\mathbf{x}_0)] - \mathrm{E}[F(\mathbf{x}_T)]}{\theta T}$$

$$\leq \frac{\sigma^2}{2\theta L|\mathcal{S}_1|} + \frac{\mathrm{E}[F(\mathbf{x}_0)] - \mathrm{E}[F(\mathbf{x}_*)]}{\theta T}$$

$$\leq \frac{\sigma^2}{2\theta L|\mathcal{S}_1|} + \frac{\Delta}{\theta T}, \qquad (24)$$

where the second inequality is due to $F(\mathbf{x}_*) = \min_{\mathbf{x} \in \mathbb{R}^d} F(\mathbf{x})$; the last inequality is due to Assumption 1 (iii).

On the other hand, similar to the proof of Theorem 2 we also have

$$\left\| \mathbf{g}_t - \nabla f(\mathbf{x}_{t+1}) + \frac{1}{\eta}(\mathbf{x}_{t+1} - \mathbf{x}_t) \right\|^2$$

$$\leq 2\|\mathbf{g}_t - \nabla f(\mathbf{x}_t)\|^2 + \frac{2(F(\mathbf{x}_t) - F(\mathbf{x}_{t+1}))}{\eta} + \left(2L^2 + \frac{1}{\eta^2} + \frac{2L}{\eta}\right)\|\mathbf{x}_{t+1} - \mathbf{x}_t\|^2,$$

By taking the expectation on both sides of above inequality, we get

$$E[\|\mathbf{g}_t - \nabla f(\mathbf{x}_{t+1}) + \frac{1}{\eta}(\mathbf{x}_{t+1} - \mathbf{x}_t)\|^2]$$

$$\leq 2E[\|\mathbf{g}_t - \nabla f(\mathbf{x}_t)\|^2] + \frac{2(E[F(\mathbf{x}_t)] - E[F(\mathbf{x}_{t+1})])}{\eta} + (2L^2 + \frac{1}{\eta^2} + \frac{2L}{\eta})E[\|\mathbf{x}_{t+1} - \mathbf{x}_t\|^2],$$

(25)

Plugging inequality (20) into inequality (25),

$$E[\|\mathbf{g}_t - \nabla f(\mathbf{x}_{t+1}) + \frac{1}{\eta}(\mathbf{x}_{t+1} - \mathbf{x}_t)\|^2]$$

$$\leq \frac{2(E[F(\mathbf{x}_t)] - E[F(\mathbf{x}_{t+1})])}{\eta} + (2L^2 + \frac{1}{\eta^2} + \frac{2L}{\eta})E[\|\mathbf{x}_{t+1} - \mathbf{x}_t\|^2]$$

$$+ 2\left( \frac{L^2}{|\mathcal{S}_2|} \sum_{i=(n_t-1)q}^{t} E[\|\mathbf{x}_{i+1} - \mathbf{x}_i\|^2] + E[\|\mathbf{g}_{(n_t-1)q} - \nabla f(\mathbf{x}_{(n_t-1)q})\|^2] \right).$$

Therefore, we have

$$\frac{2(E[F(\mathbf{x}_{t+1})] - E[F(\mathbf{x}_t)])}{\eta} + E[\|\mathbf{g}_t - \nabla f(\mathbf{x}_{t+1}) + \frac{1}{\eta}(\mathbf{x}_{t+1} - \mathbf{x}_t)\|^2]$$

$$\leq (2L^2 + \frac{1}{\eta^2} + \frac{2L}{\eta})E[\|\mathbf{x}_{t+1} - \mathbf{x}_t\|^2] + \frac{2L^2}{|\mathcal{S}_2|} \sum_{i=(n_t-1)q}^{t} E[\|\mathbf{x}_{i+1} - \mathbf{x}_i\|^2] + \frac{2\sigma^2}{|\mathcal{S}_1|}.$$

(26)

For any $t$ such that $(n_t - 1)q \leq t \leq n_t q - 1$, we take the telescoping sum of (26) over $t$ from $(n_t - 1)q$ to $t$.

$$\frac{2(E[F(\mathbf{x}_{t+1})] - E[F(\mathbf{x}_{(n_t-1)q})])}{\eta} + \sum_{j=(n_t-1)q}^{t} E[\|\mathbf{g}_j - \nabla f(\mathbf{x}_{j+1}) + \frac{1}{\eta}(\mathbf{x}_{j+1} - \mathbf{x}_j)\|^2]$$

$$\leq (2L^2 + \frac{1}{\eta^2} + \frac{2L}{\eta}) \sum_{j=(n_t-1)q}^{t} E[\|\mathbf{x}_{j+1} - \mathbf{x}_j\|^2] + \sum_{j=(n_t-1)q}^{t} \frac{2\sigma^2}{|\mathcal{S}_1|}$$

$$+ \frac{2L^2}{|\mathcal{S}_2|} \sum_{j=(n_t-1)q}^{t} \sum_{i=(n_t-1)q}^{j} E[\|\mathbf{x}_{i+1} - \mathbf{x}_i\|^2]$$

$$\leq (2L^2 + \frac{1}{\eta^2} + \frac{2L}{\eta}) \sum_{j=(n_t-1)q}^{t} E[\|\mathbf{x}_{j+1} - \mathbf{x}_j\|^2] + \sum_{j=(n_t-1)q}^{t} \frac{2\sigma^2}{|\mathcal{S}_1|}$$

$$+ \frac{2L^2}{|\mathcal{S}_2|} \sum_{j=(n_t-1)q}^{t} \sum_{i=(n_t-1)q}^{t} E[\|\mathbf{x}_{i+1} - \mathbf{x}_i\|^2]$$

$$\leq (2L^2 + \frac{1}{\eta^2} + \frac{2L}{\eta}) \sum_{j=(n_t-1)q}^{t} E[\|\mathbf{x}_{j+1} - \mathbf{x}_j\|^2] + \sum_{j=(n_t-1)q}^{t} \frac{2\sigma^2}{|\mathcal{S}_1|}$$

$$+ \frac{2qL^2}{|\mathcal{S}_2|} \sum_{i=(n_t-1)q}^{t} E[\|\mathbf{x}_{i+1} - \mathbf{x}_i\|^2]$$

$$= \gamma \sum_{j=(n_t-1)q}^{t} E[\|\mathbf{x}_{j+1} - \mathbf{x}_j\|^2] + \sum_{j=(n_t-1)q}^{t} \frac{2\sigma^2}{|\mathcal{S}_1|}.$$

where the second inequality is due to $j \le t$; the third inequality is due to $(n_t - 1)q \le t \le n_t q - 1$; $\gamma = 2L^2 + \frac{1}{\eta^2} + \frac{2L}{\eta} + \frac{2L^2 q}{|\mathcal{S}_2|}$. Therefore we have

$$\frac{2(\mathrm{E}[F(\mathbf{x}_{t+1})] - \mathrm{E}[F(\mathbf{x}_{(n_t-1)q})])}{\eta} \le \gamma \sum_{j=(n_t-1)q}^{t} \mathrm{E}[\|\mathbf{x}_{j+1} - \mathbf{x}_j\|^2] + \sum_{j=(n_t-1)q}^{t} \frac{2\sigma^2}{|\mathcal{S}_1|}$$
$$- \sum_{j=(n_t-1)q}^{t} \mathrm{E}[\|\mathbf{g}_j - \nabla f(\mathbf{x}_{j+1}) + \frac{1}{\eta}(\mathbf{x}_{j+1} - \mathbf{x}_j)\|^2].$$

Then

$$\frac{2}{\eta}(\mathrm{E}[F(\mathbf{x}_T)] - \mathrm{E}[F(\mathbf{x}_0)])$$
$$= \frac{2}{\eta}(\mathrm{E}[F(\mathbf{x}_T)] - \mathrm{E}[F(\mathbf{x}_{(n_T-1)q})] + \cdots + \mathrm{E}[F(\mathbf{x}_{2q})] - \mathrm{E}[F(\mathbf{x}_q)] + \mathrm{E}[F(\mathbf{x}_q)] - \mathrm{E}[F(\mathbf{x}_0)])$$
$$\le \gamma \sum_{j=0}^{T-1} \mathrm{E}[\|\mathbf{x}_{j+1} - \mathbf{x}_j\|^2] + \sum_{j=0}^{T-1} \frac{2\sigma^2}{|\mathcal{S}_1|} - \sum_{j=0}^{T-1} \mathrm{E}[\|\mathbf{g}_j - \nabla f(\mathbf{x}_{j+1}) + \frac{1}{\eta}(\mathbf{x}_{j+1} - \mathbf{x}_j)\|^2].$$

Dividing by $T$ on both sides of above inequality and rearranging it we have

$$\frac{1}{T} \sum_{t=0}^{T-1} \mathrm{E}[\|\mathbf{g}_t - \nabla f(\mathbf{x}_{t+1}) + \frac{1}{\eta}(\mathbf{x}_{t+1} - \mathbf{x}_t)\|^2]$$
$$\le \frac{2(\mathrm{E}[F(\mathbf{x}_0)] - \mathrm{E}[F(\mathbf{x}_T)])}{\eta T} + \gamma \frac{1}{T} \sum_{t=0}^{T-1} \mathrm{E}[\|\mathbf{x}_{t+1} - \mathbf{x}_t\|^2] + \frac{2\sigma^2}{|\mathcal{S}_1|}$$
$$\le \frac{2\Delta}{\eta T} + \gamma \frac{1}{T} \sum_{t=0}^{T-1} \mathrm{E}[\|\mathbf{x}_{t+1} - \mathbf{x}_t\|^2] + \frac{2\sigma^2}{|\mathcal{S}_1|}. \tag{27}$$

Combining above inequality with (17) and (24) and taking the expectation, we have

$$\mathrm{E}_R[\mathrm{dist}(0, \hat{\partial} F(\mathbf{x}_R))^2]$$
$$= \frac{1}{T} \sum_{t=0}^{T-1} \mathrm{E}[\|\mathbf{g}_t - \nabla f(\mathbf{x}_{t+1}) + \frac{1}{\eta}(\mathbf{x}_{t+1} - \mathbf{x}_t)\|^2]$$
$$\le \frac{2\Delta}{\eta T} + \gamma \frac{1}{T} \sum_{t=0}^{T-1} \mathrm{E}[\|\mathbf{x}_{t+1} - \mathbf{x}_t\|^2] + \frac{2\sigma^2}{|\mathcal{S}_1|}$$
$$\le \frac{2\Delta}{\eta T} + \gamma \left( \frac{\sigma^2}{2\theta L |\mathcal{S}_1|} + \frac{\Delta}{\theta T} \right) + \frac{2\sigma^2}{|\mathcal{S}_1|}$$
$$= \frac{2\theta\Delta + \gamma\eta\Delta}{\eta\theta T} + \frac{(\gamma + 4\theta L)\sigma^2}{2\theta L |\mathcal{S}_1|},$$

where $\gamma = 2L^2 + \frac{1}{\eta^2} + \frac{2L}{\eta} + \frac{2L^2 q}{|\mathcal{S}_2|}$, and $\theta = \frac{1-2\eta L}{2\eta} - \frac{Lq}{2|\mathcal{S}_2|}$. Since $q = |\mathcal{S}_2|$ and $\eta = \frac{c}{L}$ with $0 < c < \frac{1}{3}$, then $\theta = \frac{1-3\eta L}{2\eta} > 0$ and $\gamma = 4L^2 + \frac{1}{\eta^2} + \frac{2L}{\eta}$.

For the finite-sum setting, the proof can be obtained by a slight change in above analysis using the fact that

$$\mathrm{E}[\|\mathbf{g}_{(n_t-1)q} - \nabla f(\mathbf{x}_{(n_t-1)q})\|^2] = 0.$$

Then Lemma 1 will give us

$$\mathrm{E}[\|\mathbf{g}_t - \nabla f(\mathbf{x}_t)\|^2] \le \frac{L^2}{|\mathcal{S}_2|} \sum_{i=(n_t-1)q}^{t} \mathrm{E}[\|\mathbf{x}_{i+1} - \mathbf{x}_i\|^2].$$

Following the similar analysis, we will have

$$\mathrm{E}_R[\mathrm{dist}(0, \hat{\partial} F(\mathbf{x}_R))^2] \le \frac{2\theta\Delta + \gamma\eta\Delta}{\eta\theta T}.$$

$\square$

## D  Proof of Corollary 6

*Proof.* The proof uses the results in Theorem 5.

**Online setting:** The total complexity is

$$|\mathcal{S}_2|T + |\mathcal{S}_1|\left\lceil\frac{T}{q}\right\rceil \le |\mathcal{S}_2|T + |\mathcal{S}_1|\frac{T}{q} + |\mathcal{S}_1|$$

$$=\sqrt{\frac{4(\gamma + \theta L)\sigma^2}{\theta L\epsilon^2}} \cdot \frac{2(2\theta + \gamma\eta)\Delta}{\eta\theta\epsilon^2}$$

$$+ \frac{(\gamma + 4\theta L)\sigma^2}{\theta L\epsilon^2} \cdot \frac{2(2\theta + \gamma\eta)\Delta}{\eta\theta\epsilon^2} \cdot \sqrt{\frac{\theta L\epsilon^2}{4(\gamma + \theta L)\sigma^2}} + \frac{(\gamma + 4\theta L)\sigma^2}{\theta L\epsilon^2}$$

$$=O(\epsilon^{-3}).$$

**Finite-sum setting:** The proof can be obtained by a slight change in the proof of Theorem 5 using the fact that

$$\mathrm{E}[\|\mathbf{g}_{(n_t-1)q} - \nabla f(\mathbf{x}_{(n_t-1)q})\|^2] = 0.$$

Then the total complexity is

$$|\mathcal{S}_2|T + |\mathcal{S}_1|\left\lceil\frac{T}{q}\right\rceil \le |\mathcal{S}_2|T + |\mathcal{S}_1|\frac{T}{q} + |\mathcal{S}_1|$$

$$=\sqrt{n} \cdot \frac{(2\theta + \gamma\eta)\Delta}{\eta\theta\epsilon^2} + n \cdot \frac{(2\theta + \gamma\eta)\Delta}{\eta\theta\epsilon^2} \cdot \sqrt{\frac{1}{n}} + n$$

$$=O(\sqrt{n}\epsilon^{-2} + n).$$

$\square$

## E  Proof of Theorem 7

*Proof.* We first focus on the online setting. Following the similar analysis of Theorem 5 we have

$$\frac{1}{T}\sum_{t=0}^{T-1}\mathrm{E}[F(\mathbf{x}_{t+1})] - \mathrm{E}[F(\mathbf{x}_t)]$$

$$\le\frac{1}{2LT}\sum_{t=0}^{T-1}\mathrm{E}[\|\mathbf{g}_t - \nabla f(\mathbf{x}_t)\|^2] - \frac{1 - 2\eta L}{2\eta T}\sum_{t=0}^{T-1}\mathrm{E}[\|\mathbf{x}_{t+1} - \mathbf{x}_t\|^2]. \qquad (28)$$

We want to upper bound the variance term $\sum_{t=0}^{T-1}\mathrm{E}[\|\mathbf{g}_t - \nabla f(\mathbf{x}_t)\|^2]$ by using Lemma 1 of [18]. By the updates of Algorithm 3 we know it can be written as

$$\sum_{t=0}^{T-1}\mathrm{E}[\|\mathbf{g}_t - \nabla f(\mathbf{x}_t)\|^2]$$

$$=\sum_{j=0}^{b-1}\mathrm{E}[\|\mathbf{g}_j - \nabla f(\mathbf{x}_j)\|^2] + \sum_{j=b}^{3b-1}\mathrm{E}[\|\mathbf{g}_j - \nabla f(\mathbf{x}_j)\|^2] + \sum_{j=3b}^{6b-1}\mathrm{E}[\|\mathbf{g}_j - \nabla f(\mathbf{x}_j)\|^2]$$

$$+ \cdots + \sum_{j=s(s-1)b/2}^{s(s+1)b/2-1}\mathrm{E}[\|\mathbf{g}_j - \nabla f(\mathbf{x}_j)\|^2] + \cdots + \sum_{j=S(S-1)b/2}^{T-1}\mathrm{E}[\|\mathbf{g}_j - \nabla f(\mathbf{x}_j)\|^2] \qquad (29)$$

In particular, by Lemma 1, for any $t$ such that $s(s-1)b/2 \le t \le s(s+1)b/2 - 1$ in Algorithm 3, we have

$$E[\|\mathbf{g}_t - \nabla f(\mathbf{x}_t)\|^2] \le \frac{L^2}{|\mathcal{S}_{2,s}|} \sum_{i=s(s-1)b/2}^{t} E[\|\mathbf{x}_{i+1} - \mathbf{x}_i\|^2] + E[\|\mathbf{g}_{s(s-1)b/2} - \nabla f(\mathbf{x}_{s(s-1)b/2})\|^2]$$

$$\le \frac{L^2}{|\mathcal{S}_{2,s}|} \sum_{i=s(s-1)b/2}^{t} E[\|\mathbf{x}_{i+1} - \mathbf{x}_i\|^2] + \frac{\sigma^2}{|\mathcal{S}_{1,s}|}, \tag{30}$$

where the second inequality is due to Assumption 1 (ii). For any $t$ such that $s(s-1)b/2 \le t \le s(s+1)b/2 - 1$, we take the telescoping sum of (30) over $t$ from $s(s-1)b/2$ to $t$.

$$\sum_{j=s(s-1)b/2}^{t} E[\|\mathbf{g}_j - \nabla f(\mathbf{x}_j)\|^2]$$

$$\le \frac{L^2}{|\mathcal{S}_{2,s}|} \sum_{j=s(s-1)b/2}^{t} \sum_{i=s(s-1)b/2}^{j} E[\|\mathbf{x}_{i+1} - \mathbf{x}_i\|^2] + \sum_{j=s(s-1)b/2}^{t} \frac{\sigma^2}{|\mathcal{S}_{1,s}|}$$

$$\le \frac{L^2}{|\mathcal{S}_{2,s}|} \sum_{j=s(s-1)b/2}^{t} \sum_{i=s(s-1)b/2}^{t} E[\|\mathbf{x}_{i+1} - \mathbf{x}_i\|^2] + \sum_{j=s(s-1)b/2}^{t} \frac{\sigma^2}{|\mathcal{S}_{1,s}|}$$

$$\le \frac{L^2 bs}{|\mathcal{S}_{2,s}|} \sum_{i=s(s-1)b/2}^{t} E[\|\mathbf{x}_{i+1} - \mathbf{x}_i\|^2] + \sum_{j=s(s-1)b/2}^{t} \frac{\sigma^2}{|\mathcal{S}_{1,s}|}$$

$$= L^2 \sum_{j=s(s-1)b/2}^{t} E[\|\mathbf{x}_{j+1} - \mathbf{x}_j\|^2] + \sum_{j=s(s-1)b/2}^{t} \frac{\sigma^2}{b^2 s^2}, \tag{31}$$

where the second inequality is due to $j \le t$; the third inequality is due to $s(s-1)b/2 \le t \le s(s+1)b/2 - 1$; the last equality is due to $|\mathcal{S}_{1,s}| = b^2 s^2$ and $|\mathcal{S}_{2,s}| = bs$. Plugging inequality (31) into equality (29), we get

$$\sum_{t=0}^{T-1} E[\|\mathbf{g}_t - \nabla f(\mathbf{x}_t)\|^2]$$

$$\le \sum_{j=0}^{b-1} \left( L^2 E[\|\mathbf{x}_{j+1} - \mathbf{x}_j\|^2] + \frac{\sigma^2}{b^2 1^2} \right) + \sum_{j=b}^{3b-1} \left( L^2 E[\|\mathbf{x}_{j+1} - \mathbf{x}_j\|^2] + \frac{\sigma^2}{b^2 2^2} \right)$$

$$+ \cdots + \sum_{j=s(s-1)b/2}^{s(s+1)b/2-1} \left( L^2 E[\|\mathbf{x}_{j+1} - \mathbf{x}_j\|^2] + \frac{\sigma^2}{b^2 s^2} \right) + \cdots$$

$$+ \sum_{j=S(S-1)b/2}^{T-1} \left( L^2 E[\|\mathbf{x}_{j+1} - \mathbf{x}_j\|^2] + \frac{\sigma^2}{b^2 S^2} \right)$$

$$= L^2 \sum_{t=0}^{T-1} E[\|\|\mathbf{x}_{t+1} - \mathbf{x}_t\|^2] + \sum_{j=0}^{b-1} \frac{\sigma^2}{b^2 1^2} + \sum_{j=b}^{3b-1} \frac{\sigma^2}{b^2 2^2} + \cdots + \sum_{j=s(s-1)b/2}^{s(s+1)b/2-1} \frac{\sigma^2}{b^2 s^2} + \cdots$$

$$+ \sum_{j=S(S-1)b/2}^{T-1} \frac{\sigma^2}{b^2 S^2}$$

$$= L^2 \sum_{t=0}^{T-1} E[\|\|\mathbf{x}_{t+1} - \mathbf{x}_t\|^2] + \sum_{s=1}^{S} \frac{\sigma^2}{bs}. \tag{32}$$

Plugging above inequality (32) into inequality (28) we then have

$$\frac{1}{T}\sum_{t=0}^{T-1}\mathrm{E}[F(\mathbf{x}_{t+1})] - \mathrm{E}[F(\mathbf{x}_t)]$$

$$\leq \frac{1}{2L}\frac{1}{T}\left(L^2\sum_{t=0}^{T-1}\mathrm{E}[\|\|\mathbf{x}_{t+1}-\mathbf{x}_t\|^2] + \sum_{s=1}^{S}\frac{\sigma^2}{bs}\right) - \frac{1-2\eta L}{2\eta}\frac{1}{T}\sum_{t=0}^{T-1}\mathrm{E}[\|\mathbf{x}_{t+1}-\mathbf{x}_t\|^2]. \qquad (33)$$

Rearranging the inequality (33), we know

$$\frac{1}{T}\sum_{t=0}^{T-1}\mathrm{E}[\|\mathbf{x}_{t+1}-\mathbf{x}_t\|^2] \leq \frac{\mathrm{E}[F(\mathbf{x}_0)]-\mathrm{E}[F(\mathbf{x}_T)]}{\theta T} + \frac{1}{2\theta LT}\sum_{s=1}^{S}\frac{\sigma^2}{bs} \leq \frac{\Delta}{\theta T} + \frac{1}{2\theta LT}\sum_{s=1}^{S}\frac{\sigma^2}{bs}, \qquad (34)$$

where $\theta := \frac{1-3\eta L}{2\eta} > 0$.

On the other hand, similar to the proof of Theorem 5 by (25) we also have

$$\frac{1}{T}\sum_{t=0}^{T-1}\mathrm{E}[\|\mathbf{g}_t - \nabla f(\mathbf{x}_{t+1}) + \frac{1}{\eta}(\mathbf{x}_{t+1}-\mathbf{x}_t)\|^2]$$

$$\leq 2\frac{1}{T}\sum_{t=0}^{T-1}\mathrm{E}[\|\mathbf{g}_t - \nabla f(\mathbf{x}_t)\|^2] + \frac{1}{T}\sum_{t=0}^{T-1}\frac{2(\mathrm{E}[F(\mathbf{x}_t)]-\mathrm{E}[F(\mathbf{x}_{t+1})])}{\eta}$$

$$+ (2L^2 + \frac{1}{\eta^2} + \frac{2L}{\eta})\frac{1}{T}\sum_{t=0}^{T-1}\mathrm{E}[\|\mathbf{x}_{t+1}-\mathbf{x}_t\|^2]$$

$$\leq 2\frac{1}{T}\sum_{t=0}^{T-1}\mathrm{E}[\|\mathbf{g}_t - \nabla f(\mathbf{x}_t)\|^2] + \frac{2\Delta}{\eta T} + (2L^2 + \frac{1}{\eta^2} + \frac{2L}{\eta})\frac{1}{T}\sum_{t=0}^{T-1}\mathrm{E}[\|\mathbf{x}_{t+1}-\mathbf{x}_t\|^2]. \qquad (35)$$

Plugging inequality (32) into inequality (35),

$$\frac{1}{T}\sum_{t=0}^{T-1}\mathrm{E}[\|\mathbf{g}_t - \nabla f(\mathbf{x}_{t+1}) + \frac{1}{\eta}(\mathbf{x}_{t+1}-\mathbf{x}_t)\|^2]$$

$$\leq 2\frac{L^2\sum_{t=0}^{T-1}\mathrm{E}[\|\|\mathbf{x}_{t+1}-\mathbf{x}_t\|^2] + \sum_{s=1}^{S}\frac{\sigma^2}{bs}}{T} + \frac{2\Delta}{\eta T} + (2L^2 + \frac{1}{\eta^2} + \frac{2L}{\eta})\frac{1}{T}\sum_{t=0}^{T-1}\mathrm{E}[\|\mathbf{x}_{t+1}-\mathbf{x}_t\|^2]$$

$$= \frac{2}{T}\sum_{s=1}^{S}\frac{\sigma^2}{bs} + \frac{2\Delta}{\eta T} + (4L^2 + \frac{1}{\eta^2} + \frac{2L}{\eta})\frac{1}{T}\sum_{t=0}^{T-1}\mathrm{E}[\|\mathbf{x}_{t+1}-\mathbf{x}_t\|^2]$$

$$\leq \frac{2}{T}\sum_{s=1}^{S}\frac{\sigma^2}{bs} + \frac{2\Delta}{\eta T} + (4L^2 + \frac{1}{\eta^2} + \frac{2L}{\eta})\left(\frac{\Delta}{\theta T} + \frac{1}{2\theta LT}\sum_{s=1}^{S}\frac{\sigma^2}{bs}\right)$$

$$= \frac{(2\theta+\gamma\eta)\Delta}{\theta\eta T} + \frac{4\theta L+\gamma}{2\theta LT}\sum_{s=1}^{S}\frac{\sigma^2}{bs}, \qquad (36)$$

where $\gamma = 4L^2 + \frac{1}{\eta^2} + \frac{2L}{\eta}$, the last inequality is due to (34). Combining above inequality with the fact that $\nabla f(\mathbf{x}_{t+1}) - \mathbf{g}_t - \frac{1}{\eta}(\mathbf{x}_{t+1}-\mathbf{x}_t) \in \nabla f(\mathbf{x}_{t+1}) + \hat{\partial} r(\mathbf{x}_{t+1}) = \hat{\partial} F(\mathbf{x}_{t+1})$ and taking the

expectation, we have

$$\mathrm{E}_R[\mathrm{dist}(0, \hat{\partial}F(\mathbf{x}_R))^2]$$

$$\leq \frac{1}{T}\sum_{t=0}^{T-1}\mathrm{E}[\|\mathbf{g}_t - \nabla f(\mathbf{x}_{t+1}) + \frac{1}{\eta}(\mathbf{x}_{t+1} - \mathbf{x}_t)\|^2]$$

$$\leq \frac{(2\theta + \gamma\eta)\Delta}{\theta\eta T} + \frac{4\theta L + \gamma}{2\theta LT}\sum_{s=1}^{S}\frac{\sigma^2}{bs}$$

$$\leq \frac{(2\theta + \gamma\eta)\Delta}{\theta\eta T} + \frac{(4\theta L + \gamma)\sigma^2(\log(S) + 1)}{2b\theta LT}$$

$$\leq \frac{(2\theta + \gamma\eta)\Delta}{\theta\eta T} + \frac{(4\theta L + \gamma)\sigma^2(\frac{1}{2}\log(2T/b) + 1)}{2b\theta LT},$$

where the last second inequality is due to $\sum_{s=1}^{S}\frac{1}{s} \leq \log(S) + 1$; the last inequality is due to $S \leq \sqrt{S(S+1)} = \sqrt{\frac{2T}{b}}$. Since $\eta = \frac{c}{L}$ with $0 < c < \frac{1}{3}$, then $\theta = \frac{1-3\eta L}{2\eta} > 0$.

Similarly, the proof for the finite-sum setting can be obtained by a slight change in above analysis using the fact that

$$\mathrm{E}[\|\mathbf{g}_{(n_t-1)q} - \nabla f(\mathbf{x}_{(n_t-1)q})\|^2] = 0.$$

in Lemma 1. Following the similar analysis, we will have

$$\mathrm{E}_R[\mathrm{dist}(0, \hat{\partial}F(\mathbf{x}_R))^2] \leq \frac{2\theta\Delta + \gamma\eta\Delta}{\eta\theta T}.$$

$\square$

# F   Heuristic SGD for learning a quantized model

We present a popular heuristic SGD approach in deep learning for learning a quantized model [36] in the following algorithm. The step size is reduced by half at each epoch in the experiments.

---
**Algorithm 5** Heuristic SGD for learning a quantized model [36]
---
1: **Initialize**: $\mathbf{x}_0 \in \mathbb{R}^d$, $\eta_0 = \eta$ and $\hat{\mathbf{x}}_0 = P_\Omega(\mathbf{x}_0)$ is the quantized model
2: **for** $t = 0, 1, \ldots, T-1$ **do**
3:    $\mathbf{x}_{t+1} = \mathbf{x}_t - \eta\nabla f(\hat{\mathbf{x}}_t; \xi_t)$, where $\hat{\mathbf{x}}_t = P_\Omega(\mathbf{x}_t)$ is the quantized model
4:    **if** $\mathrm{mod}(t, n) == 0$ **then**
5:        $\eta = \eta/2$
6:    **end if**
7: **end for**
8: **Output:** $\mathbf{x}_R$, where $R$ is uniformly sampled from $\{1, \ldots, T\}$.
---