[Reviews · NeurIPS 2019]

Reviewer 1



1.One of the applications for the proposed algorithm may be the case with the zero norm. However, for the zero norm, extensive theoretical analyses have been done, such as matching pursuit, IHT, and StoIHT. Some of them even guarantee perfect support recovery. Comparing with those algorithms, what are the benefits of the newly-proposed algorithms? Here, first-order stationary points may be far away from the optimal solution when the sparsity regularizer is imposed. It might be interesting to know the property of support recovery for the proposed algorithms. Or in a more general case, what is the behavior related to the convergence of x_t to the “optimal x^*”? 2. Based on Theorem 5, the proposed algorithm 3 will not converge to a first-order stationary point, since $(\gamma + 4\theta L ) sigma^2 / 2 \theta L |S_1| $ is dependent on batch size |S_1| and |S_1| is upper bounded by data size n ( at least in the finite-sum setting ). In order to make (\gamma + 4\theta L ) sigma^2 / 2 \theta L |S_1| \leq O(\epsilon), |S_1| may need to be larger than n, which is not possible. At least from Theorem 5, this reviewer has problems understanding why it is true for the finite-sum setting in Corrollary 6, which may need further explanation. 3. Please give the definition of \tilda( O) and O in table 1. 4, line 76 typo: mataching -> matching

Reviewer 2



This paper provides the first non-asymptotic analysis for stochastic proximal gradient algorithms in the fully non-smooth non-convex setting. So far only convergence results in the deterministic case, and recently in the stochastic case, have been obtained. The rates derived are optimal. No regularization or smoothing of the objective is performed (such as reformulating an auxiliary function like the Moreau envelope), but rather a direct proximal sub gradient step is performed at every stage. The drawback of this approach is that no potential function is available, making it difficult how to measure progress of the algorithm. Moreover, Theorem 2 shows that the distance measure to the set of stationary points is given by a sum of variances of the stochastic oracle. As a consequence, it has to be assumed that the stochastic oracle's noise is uniformly bounded, which is a quite strong assumption. Specific comments: -) line 23: If r(x) is assumed to be LSC it can be non-smooth as well. There is no need to emphasize this again here. -) line 76: typo in „matching“. -) line 87: It is a bit unfortunate to use the same letter for a data sample and a generic set. I would recommend introducing different notation to distinguish them. -) I don’t agree that eq. (2) is a special case of (1). Mathematically speaking this is correct of course, but the interpretation is usually different. In Stochastic optimization our goal is to solve problem (1) and problem (2) would represent a sample average stochastic approximation for this problem. -) line 106: I don’t think that argmin needs an explanation. -) line 110: two times „the“ -) The iteration complexity of Theorem 1 depends on the quantity Delta. It seems to be hard to have a good guess for this quantity, unless we restrict the analysis to an a-priori known compact set. How do you justify this assumption? -) Proof of Theorem 2: In line 421, instead of what is written, you should use the Fenchel-Young inequality to get to the bound (16), i.e a.b<(1/2L)a^{2}+(L/2)b^{2}. Same remark applies to line 443. Line 429 is just the Pythagoras Theorem.

Reviewer 3



For neural network compression, it is common to train sparse models with L0 projections, as well as quantized models. It is significant to analyze and propose new algorithms for such training. The algorithms are limited in practice because the learning rate is constant. The practical fix of increasing the batch size throughout training is significantly less convenient than decreasing the learning rate. For the convergence analysis, the main trick is to exploit the optimality conditions from the proximal update, and I found the proof sketch in Section 2.1 instructive. *Update* I have read the author response and other reviews, and I am keeping my review the same. I think my point about the practicality is still true, but this is only a minor comment. The authors plan to define the practical algorithm in a way that is more clear.

[Author Response · NeurIPS 2019]

We thank all reviewers for their comments and suggestions!

**Reviewer 1: Q1. About the benefits of the newly-proposed algorithms.**
A. First of all, we emphasize that our goal is not to develop an algorithm for solving $\ell_0$ norm constrained problem
and prove an exact recovery result, but rather to analyze stochastic proximal gradient (SPG) for handling a general
non-convex regularizer under minimal assumptions about the problem. The benefits of the newly-proposed algorithms
is that **it is applicable to a much broader family of problems**. First, we are not restricted to $\ell_0$ norm constrained or
regularized problems. As long as the regularizer's proximal mapping can be efficiently computed, our algorithms and
their convergence guarantee are applicable (c.f. our experiments for learning with quantization). Second, we **do not**
**impose stringent condition** on the data matrix or the loss function, such as restricted isometry property or restricted
eigenvalue or restricted strong convexity that is typical for traditional sparse recovery algorithms (e.g., IHT, StoIHT).
Third, our results are applicable to any smooth loss functions even if they are non-convex, while most previous results
are restricted to convex loss. We believe adding such stringent conditions one could derive much stronger result of SPG
for $\ell_0$ norm constrained problems following existing works (e.g., [R1,R2]). But it is not the focus of this paper.

**Reviewer 1: Q2. About Theorem 5 and convergence.**
A. Thanks for this great question! Please note that this is not an error. We will make the statement of Theorem 5 more
clear in the revision (somehow the current upper bound in Thm. 5 is to capture the online setting). In fact, for the
finite-sum setting, the second term $(\gamma + 4\theta L)\sigma^2/(2\theta L|S_1|)$ will disappear in the upper bound since it is caused by the
variance of stochastic gradient $\nabla f_{S_1}(\mathbf{x}_t)$ (c.f. Line 440 of supplement). We have briefly explained in the proof of
Corollary 6 for the finite-sum setting (c.f. Line 478 of supplement) and will add more details. We will present Theorem
5 in a better way by considering the online and finite-sum setting separately. For the online setting, the current bound
holds without any change, for the finite-sum setting the upper bound only includes the first term. Thanks again!

**Reviewer 2: Q. How to justify the Assumption 1 (ii)?**
A. This assumption is quite standard and has been used in many non-convex literatures (see references [18, 19, 29, 31,
35, 41]). As long as the objective function is lower bounded, the assumption holds without assuming a compact domain.
In most machine learning applications the objective function is non-negative, i.e., $F(\mathbf{x}) \geq 0$. Hence, one can simply set
$\Delta = F(\mathbf{x}_0)$.

**Reviewer 2: Specific Comments and Improvements.**
A. We thank the reviewer for all comments. We will improve the paper following on the reviewer's comments and add
more discussion on the bounded variance assumption in connection with [29]. Thanks for the positive rating!

**Reviewer 3: Q. About the constant learning rate with a large mini-batch size vs decreasing learning rate with a**
**small mini-batch size.**
A. While we agree with the reviewer that an algorithm with a decreasing learning rate and small mini-batch size is
interesting, it might be unfair to say that an algorithm with large mini-batch size and constant learning rate is not
practical. At least, in the distributed setting it is more natural to consider a large mini-batch size rather than a small
batch size [R3]. Indeed, we have presented a variant with an increasing sequence of mini-batch sizes rather than a
large mini-batch size from the beginning. It is still an open problem to prove the **non-asymptotic convergence** of SPG
without using a large mini-batch size for a non-convex regularized problem (An asymptotic analysis of SPG without a
large batch size is presented in [15]).

**Reviewer 3: Improvements.**
A. We will formally define the practical algorithm. Thanks for the positive rating!

**Reference:**

[R1]. Linear Convergence of Stochastic Iterative Greedy Algorithms with Sparse Constraints. Nguyen et al. 2014.

[R2]. On Iterative Hard Thresholding Methods for High-dimensional M-Estimation. Jain et al, 2014.

[R3]. Accurate, Large Minibatch SGD: Training ImageNet in 1 Hour. Goyal et al. 2017.


[Meta-Review · NeurIPS 2019]

All the reviewers agree that the paper presents an interesting result and is nicely written. Please incorporate reviewers' feedback. Congratulations on a nice result.